# Virtual birefringence imaging and histological staining of amyloid deposits in label-free tissue using autofluorescence microscopy and deep learning

Xilin Yang[1,2,3], Bijie Bai[1,2,3], Yijie Zhang[1,2,3], Musa Aydin [1,4], Yuzhu Li[1,2,3], Sahan Yoruc Selcuk[1,2,3], Paloma Casteleiro Costa [1,2,3], Zhen Guo[1], Gregory A. Fishbein [5], Karine Atlan[6], William Dean Wallace [7], Nir Pillar[1,2,3] ✉ & Aydogan Ozcan [1,2,3,8] ✉

Systemic amyloidosis involves the deposition of misfolded proteins in organs/tissues, leading to progressive organ dysfunction and failure. Congo red is the gold-standard chemical stain for visualizing amyloid deposits in tissue, showing birefringence under polarization microscopy. However, Congo red staining is tedious and costly to perform, and prone to false diagnoses due to variations in amyloid amount, staining quality and manual examination of tissue under a polarization microscope. We report virtual birefringence imaging and virtual Congo red staining of label-free human tissue to show that a single neural network can transform autofluorescence images of label-free tissue into brightfield and polarized microscopy images, matching their histochemically stained versions. Blind testing with quantitative metrics and pathologist evaluations on cardiac tissue showed that our virtually stained polarization and brightfield images highlight amyloid patterns in a consistent manner, mitigating challenges due to variations in chemical staining quality and manual imaging processes in the clinical workflow.

Systemic amyloidosis is a heterogeneous group of disorders characterized by the deposition of abnormally folded proteins in tissue. The clinical picture of systemic amyloidosis is not specific, with profound fatigue, weight loss, and edema being the common presenting symptoms[1]. The real prevalence of systemic amyloidosis is not known[2,3]. A retrospective evaluation of kidney biopsies suggests that amyloidosis is not as rare as it is thought to be, accounting for ~43% of nephrotic proteinuria above age 60[4]. In another study, 31% of the patients with multiple myeloma had confirmed evidence of systemic amyloidosis[5]. Cardiac amyloid deposition, causing infiltrative/restrictive cardiomyopathy, is the leading cause of morbidity and mortality in systemic amyloidosis, regardless of the underlying pathogenesis of amyloid production[6]. Similar to other organs involved with amyloidosis, cardiac amyloidosis remains substantially underdiagnosed, and it is advised to test for cardiac amyloidosis presence during the initial work-up of all patients ≥65 years old hospitalized with heart failure[7].

[1]Electrical and Computer Engineering Department, University of California, Los Angeles, CA 90095, USA. [2]Bioengineering Department, University of California, Los Angeles, CA 90095, USA. [3]California NanoSystems Institute (CNSI), University of California, Los Angeles, CA 90095, USA. [4]Department of Computer Engineering, Fatih Sultan Mehmet Vakif University, Istanbul 34038, Turkey. [5]Department of Pathology and Laboratory Medicine, David Geffen School of Medicine at the University of California, Los Angeles, CA 90095, USA. [6]Department of Pathology, Hadassah Hebrew University Medical Center, Jerusalem 91120, Israel. [7]Department of Pathology, Keck School of Medicine, University of Southern California, Los Angeles, CA 90033, USA. [8]Department of Surgery, University of California, Los Angeles, CA 90095, USA. ✉e-mail: npillar@g.ucla.edu; ozcan@ucla.edu

Early diagnosis of systemic amyloidosis is essential to reducing morbidity and mortality of the disease. A prompt intervention following early-stage amyloid detection may save patients from extensive and irreversible tissue damage. In addition, a definitive diagnosis has become increasingly important since a number of impactful treatment options have developed[8].

Diagnosis of amyloidosis is usually based on the demonstration of amyloid deposits in a tissue biopsy. Cardiac biopsy provides the most definitive diagnostic evidence in amyloid cardiomyopathy, and endomyocardial biopsy was shown to be a safe and relatively simple procedure[9]. Congo red is considered the gold standard stain used in the vast majority of histopathology laboratories to identify amyloid in tissues. When tested under cross-polarized light microscopy, Congo red-stained amyloid areas demonstrate birefringence, which is considered a specific feature of amyloidosis. However, the traditional workflow (as depicted in Fig. 1a) has several drawbacks. Congo red staining analysis exhibits inter-observer variability, partly attributed to the challenging nature of Congo red staining[10]. In addition to a standard brightfield microscopy evaluation, visualization under polarized light microscopy is needed to examine the presence of birefringence. The quality of the examining clinical-grade microscope for highlighting amyloid birefringence can potentially limit the accuracy of pathologist evaluations, leading to an increase in both false-negative and false-positive results[11]. A false negative tissue pathology report can mislead diagnosticians and lead to the exclusion of the amyloidosis

diagnosis, often without reconsideration among the differential diagnoses. Recent reports showed that the median time from the symptom onset to amyloidosis diagnosis was ~2 years. Additionally, nearly a third of patients reported seeing at least 5 physicians before receiving a diagnosis of amyloidosis[12].

These technical challenges of Congo red staining and diagnostic inspection under polarized light microscopy also pose a barrier to the broader adoption of digital pathology[13,14]. There is currently no clinically approved digital pathology slide scanner with the required polarization imaging components, and the existing scanning polarization imagers are limited to well-resourced settings, mostly for research use[15].

Recently, deep learning-based technologies have introduced transformative opportunities to biomedical research and clinical diagnostics[14,16] using deep neural networks (DNNs) to learn intrinsic structures in large datasets[17]. A notable application of this is the use of deep learning for virtual histological staining of label-free tissue samples[18–27]. In this technique, a deep convolutional neural network (CNN) is trained to computationally stain microscopic images of unstained (label-free) tissue sections, matching their histologically stained counterparts. This computational approach aims to circumvent some of the challenges associated with traditional histochemical staining. Multiple research groups and institutions explored this deep learning-based virtual staining technique and successfully utilized it for digitally generating a wide range of routinely used stains, both

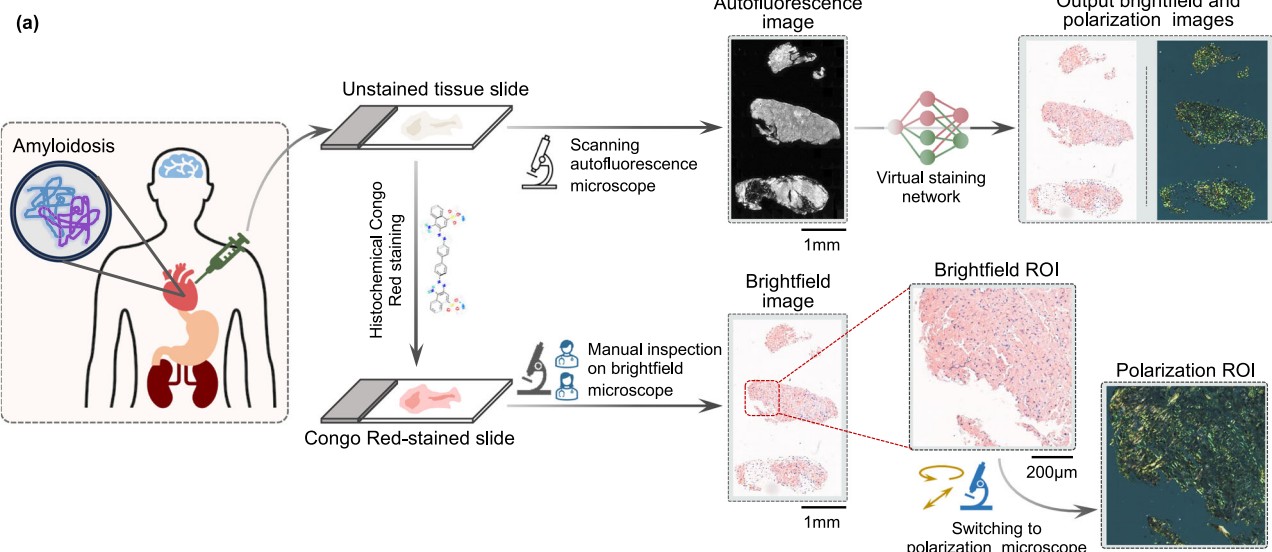

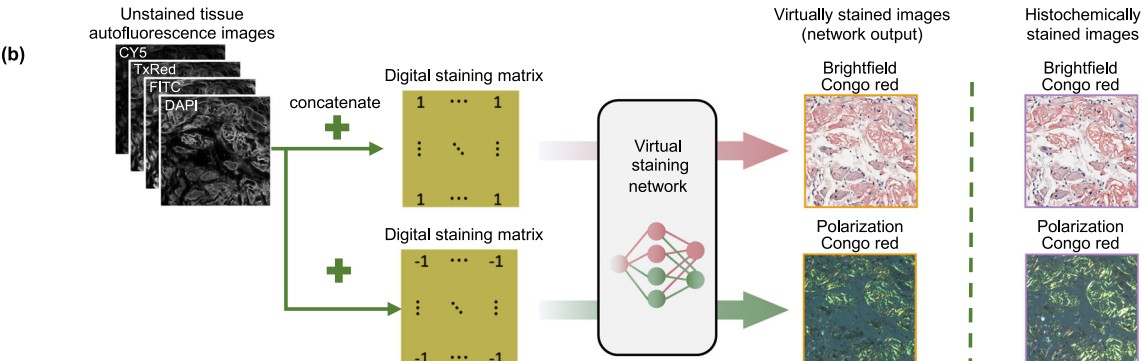

**Fig. 1 | Virtual tissue staining and birefringence imaging of label-free amyloid deposits. a** The virtual staining schematic pipeline and its comparison to clinical workflow. **b** Virtual staining network and digital staining matrix framework to generate two output modalities (brightfield and birefringence channels). Chemical structure of Congo red obtained from https://pubchem.ncbi.nlm.nih.gov/compound/11313#section=2D-Structure.

histological[18,28–30], immunohistochemical[21,31] and fluorescent stains[32]. Such deep-learning-based transformations can also be applied from one stain type to another or to blend several stains simultaneously[20,33–35], utilizing existing stained tissue images to provide more information that could help with diagnosis.

Here, we report a deep learning-based virtual tissue staining technique designed to digitally label amyloid deposits in unstained, label-free tissue sections (as illustrated in Fig. 1). This technique achieves autofluorescence to birefringence image transformations along with virtual Congo red staining of label-free tissue. It employs conditional generative adversarial networks[36] (cGANs) to rapidly and digitally convert autofluorescence microscopy images of unstained tissue slides into virtually stained birefringence and brightfield images, closely resembling the corresponding images of the histochemically stained samples, helping the identification of amyloid deposits in label-free tissue slides. We designed our deep learning model to simultaneously learn the cross-modality image transformations for the two output modalities within a single neural network, performing both autofluorescence-to-birefringence and autofluorescence-to-brightfield image transformations using a digital staining matrix (DSM), illustrated in Fig. 1b, which is an additional channel concatenated to the input autofluorescence microscopy images, indicating the desired output modality (birefringence vs. brightfield). An auxiliary registration module was also integrated into the cGAN in the training process to mitigate spatial misalignments in the training data by learning to register the output and the ground truth histochemically stained images[37]. After a one-time training phase, when given a new label-free test sample, the virtual staining network successfully generated microscopic images of the Congo red-stained tissue in both brightfield and birefringence channels. These digitally generated, virtually stained images were then stitched into whole slide images (WSI) and examined by pathologists using a customized multi-modality WSI viewer, offering the flexibility to swiftly toggle between the brightfield and polarization/birefringence views; also see Supplementary Fig. 1 for an overview of this process. Our methodology's effectiveness was affirmed by three board-certified pathologists attesting to the non-inferior quality of the images generated by our network model compared to histochemically stained images, demonstrating a high degree of concordance with the ground truth. Validated by a group of pathologists, our approach effectively bypasses the limitations of traditional histological staining workflow manually performed by histotechnologists and also eliminates the need for tedious polarization imaging with specialized optical components, facilitating a faster and more reliable diagnosis of amyloidosis.

## Results

### Label-free virtual birefringence imaging and amyloid staining
We demonstrated our virtual birefringence imaging and virtual Congo red staining technique by training a deep-learning model on a dataset comprising label-free tissue sections with a total of 386 and 65 training/validation and testing image patches, respectively, where each image patch had 2048 × 2048 pixels, all obtained from eight distinct patients. To obtain this dataset, we imaged unlabeled/label-free tissue sections with a 4-channel autofluorescence microscope and then sent these label-free slides for standard histochemical Congo red staining (for ground truth generation). The histochemically stained tissue slides were then imaged using a standard brightfield microscope scanner as well as a polarization microscope (see the Methods section). Manual fine-tuning of the polarizer and analyzer settings was conducted by a board-certified pathologist to ensure the quality of the captured birefringence patterns that served as our ground truth for the polarization channel. Following the image data acquisition, an image registration process was performed in the training phase to spatially register the brightfield and the polarization images to label-

free autofluorescence images to mitigate potential image misalignments[20,38]. The total data size for all the training, validation and testing images amounts to ~40 GB. For more details on image dataset acquisition and preprocessing of data, refer to the Method section.

Our deep learning model was composed of three sub-modules: (1) a generator designed to learn the two necessary cross-modality image transformations, i.e., autofluorescence-to-birefringence and autofluorescence-to-brightfield imaging, (2) a discriminator that engages in adversarial learning to differentiate between the output and ground truth images, thereby aiding the generator during the training process, and (3) an image registration module tasked with aligning the output images with the ground truth, which helps to mitigate residual misalignments within the dataset. The image transformations from autofluorescence to brightfield and birefringence modalities were learned within a single neural network model. To determine which output (brightfield or birefringence) is desired, a DSM was concatenated to the input. This DSM, matching the pixel count and shape of the input images, determines the output modality on a per-pixel basis with "1" indicating brightfield and "−1" indicating polarization/birefringence channel (Fig. 1b). During the training, we mixed the target images from both modalities and concatenated the corresponding DSM to the autofluorescence input images.

After the model convergence, we conducted a blind evaluation of our model using 65 test images (each with 2048×2048 pixels) obtained from two previously unseen patients. During the model testing phase, brightfield and birefringence output images were naturally aligned with the corresponding input autofluorescence images. For both the brightfield and polarization channels, the predicted virtual images exhibited a high degree of agreement with the ground truth images, as demonstrated in Fig. 2, which showcases side-by-side visual comparisons between the virtually stained images produced by our deep learning model alongside the corresponding histochemically stained ground-truth images. This figure includes two representative slides from distinct patients. Specifically, in Fig. 2a, the right side displays zoomed-in sections of the histochemically stained brightfield images, revealing regions of interest (ROIs) with a pinkish hue indicative of congophilic areas. In the corresponding polarization images, these regions exhibit an apple-green birefringence, characteristic of amyloid deposits. The virtually created images of these same regions also manifest the same morphology of amyloid presence, affirming that our inference model has effectively learned to transform autofluorescence label-free images into both brightfield and birefringence images, matching the histochemically stained counterparts, presenting an accurate appearance of amyloid deposits. The upward-pointing arrows with a blue outline highlight amyloid deposition between cardiac myocytes, while the right-pointing, yellow-outlined arrows highlight areas devoid of amyloid deposits. In Fig. 2b, a similar comparative analysis is shown for another patient. In this case, the histochemically stained images display a specific region, labeled as ROI3, which is an area without any amyloid deposits ("negative region"). No congophilic features can be identified in the brightfield image, and no apple-green birefringence is seen in the polarization channel. The upward-pointing, orange-outlined arrows denote amyloid deposition within blood vessels. As indicated by the WSIs, zoomed-in regions and arrow-pointed specific areas, our virtual staining model correctly replicated these morphological characteristics, aligning closely with the histochemically stained ground truth images, without introducing false positive staining.

### Pathologist evaluations and performance quantification
To further assess the effectiveness of our virtual staining approach, three board-certified pathologists were engaged to blindly evaluate the quality of histochemically stained and virtually stained images. The evaluation comprised two distinct parts: 1) assessing the image quality

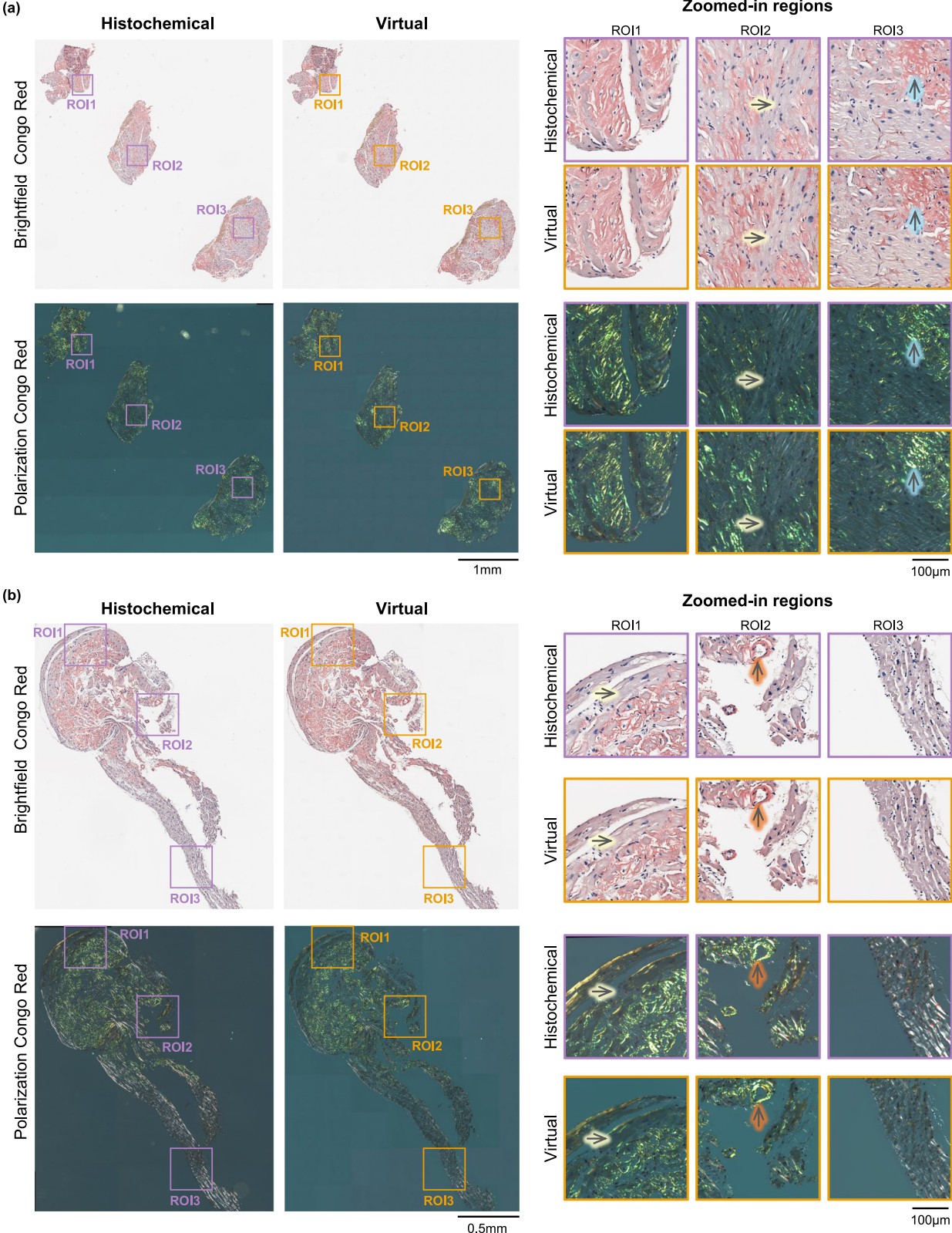

**Fig. 2 | Blind testing results of virtual birefringence imaging and histological staining of amyloid deposits.** Two whole slide samples along with zoomed-in regions are shown in (**a**, **b**). The upward-pointing arrows with a blue outline highlight amyloid deposition between cardiac myocytes, while the right-pointing, yellow-outlined arrows highlight areas devoid of amyloid deposits. The upward-pointing, orange-outlined arrows denote amyloid deposition within blood vessels. The images were generated from a single neural network inference, which is deterministic for a given input.

of brightfield Congo-red stain and 2) evaluating the appearance and quality of amyloid deposits using large, bundled patch images of the polarization and brightfield channels. The first part involved examining small patches, solely from brightfield modality, which were randomly cropped from histochemically stained and virtually stained images without overlapping field-of-views (FOVs); see Supplementary Fig. 2a). A total of 163 test image patches (each with 1024 × 1024 pixels) underwent random augmentations, were shuffled and then presented to the pathologists in a blinded manner. The expert evaluations concentrated on four primary metrics: stain quality of nuclei (M1), cytoplasm (M2), and extracellular space (M3), as well as the staining contrast of congophilic areas (M4). The first three metrics (M1–M3) are standard for evaluating stained tissue images, while the last one (M4) is unique to Congo red staining, clinically relevant for amyloidosis diagnosis. For all metrics, the grading scale ranged from 1 to 4, where 4 represents "perfect", 3 represents "very good", 2 represents "acceptable", and 1 represents "unacceptable" quality. Figure 3a visualizes the results using violin plots to compare the scores given to histochemically stained and virtually stained images. The distributions of these plots show no major differences for any of the evaluation metrics. The mean values for each pathologist (P1–P3) are plotted in Fig. 3b with an error bar representing the standard deviations (also see

Supplementary Table 1). For the first three metrics (M1–M2–M3), slightly higher scores are given to the histochemical images compared to the virtually stained images, with a mean difference of 0.229 (5.72%), 0.334 (8.35%), and 0.114 (2.85%), averaging across all pathologists and all patches–out of a scale of 4. However, for M4, which is the most relevant for Congo red staining, the difference in the mean pathologist scores falls to a negligible level of -0.067 (1.67%) out of 4. Overall, the stain quality scores corresponding to the virtual and histochemical staining are closely matched, each falling within their respective standard deviations, as shown in Fig. 3. Additional image FOVs supporting this conclusion can be found in Supplementary Fig. 3. The brightfield images of paired histochemical and virtually stained FOVs were further scored after a two-month washout period. These results, summarized in Supplementary Fig. 4, further support that the virtual brightfield images generated by our model remain satisfactory and consistent.

The second part of the expert evaluations focused on the analysis of birefringence appearance and image quality using larger FOVs and included bundled brightfield and polarization images. The same FOVs from histochemically stained and virtually stained images were included with different image augmentations (Supplementary Fig. 2b). This evaluation incorporated three distinct metrics (M5-M7):

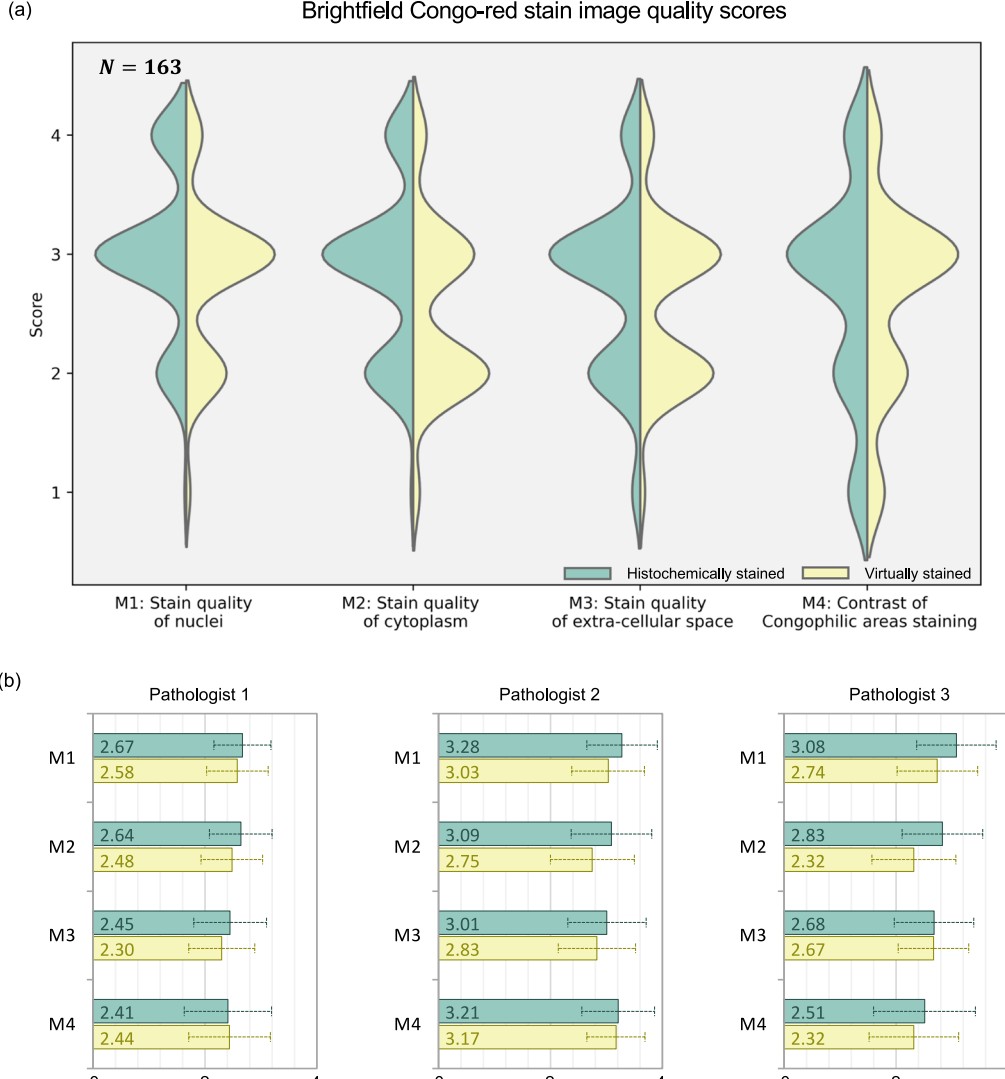

**Fig. 3 | Pathologists' blind evaluation of brightfield Congo-red stain image quality (virtually stained vs. histochemically stained).** A total of 163 image patches are evaluated, each by 3 pathologists. The scores are given for four metrics where their distributions are shown in violin plots in (**a**). The mean and standard deviation values (as error bars) are plotted for each individual pathologist in part (**b**). Source data of pathologists' scores are provided as a Source Data file.

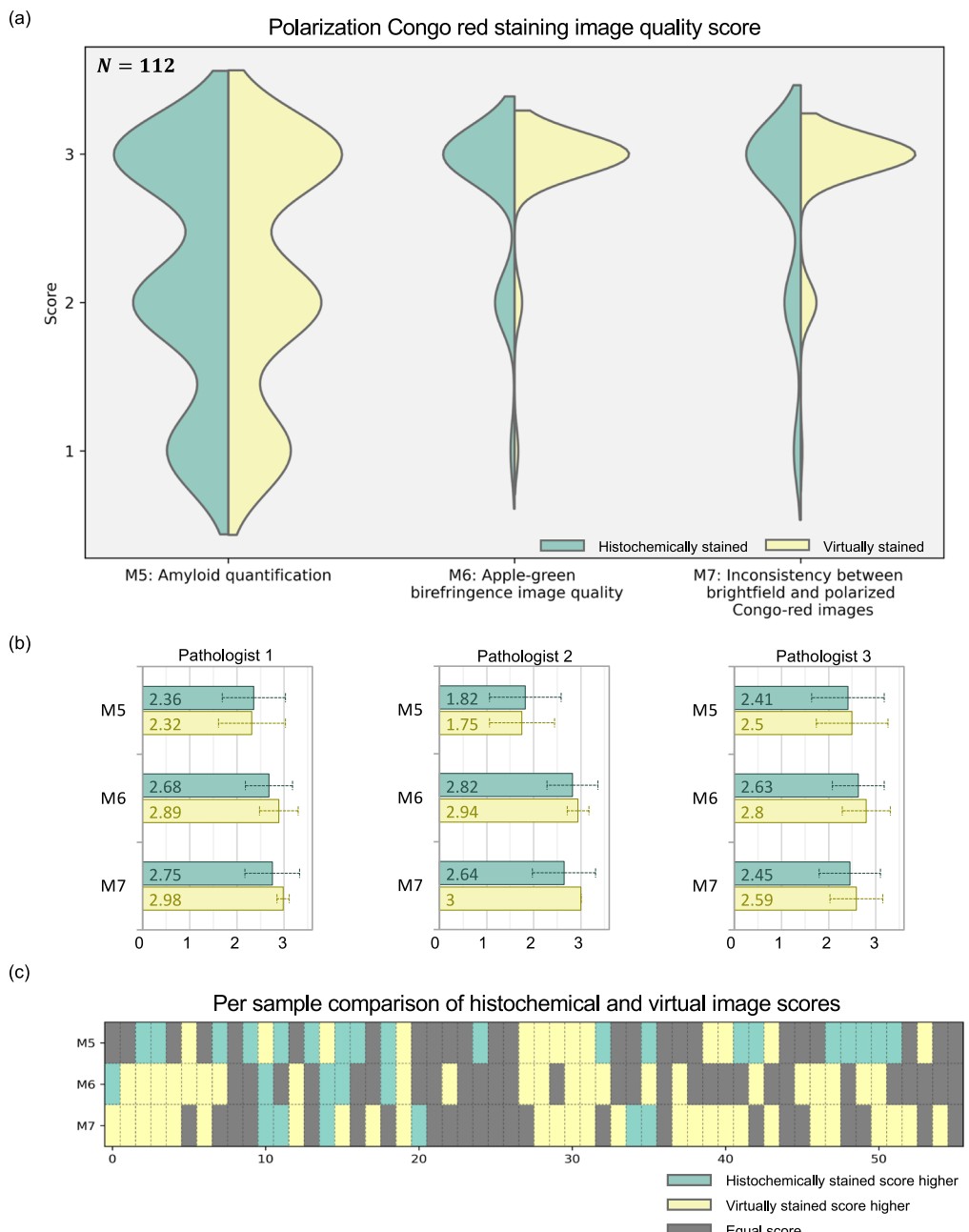

**Fig. 4 | Pathologists' blind evaluation of birefringence images of Congo red staining (virtually stained vs. histochemically stained).** A total of 112 images (56 histochemical-virtual pairs) underwent blind evaluation by three pathologists. Image quality was assessed across four metrics. Violin plots in (**a**) illustrate the distribution of scores for each metric. Panel (**b**) presents the mean and standard deviation values (as error bars) for each pathologist. Pairwise comparisons of histochemical and virtual image scores are shown in (**c**). Source data of pathologists' scores are provided as a Source Data file.

amyloid quantification (M5), the quality of birefringence appearance (M6), and the consistency between brightfield and polarization images (M7). The grading scale ranges from 1 to 3 and is different for each metric: percentage for amyloid quantification with one-third spacing (i.e., 1 = 100%); very good, moderate, and non-diagnostic for apple-green birefringence image quality; categories of low, medium, and high for inconsistency between brightfield and polarized Congo red images (lower scores indicate higher quality with less inconsistent features). A detailed description of M5-M7 scoring can be found in the Methods section with some visual examples reported in Supplementary Fig. 5. The pathologists conducted their evaluations using a custom image viewer that offered a user-friendly interface, enabling easy toggling between the brightfield and polarization

images of the same FOV and detailed examination of tissue areas of interest (refer to the Methods). Prior to the evaluation, the pathologists were familiarized with the examination process through a brief tutorial that utilized scored examples from the training dataset. Figure 4a presents a violin plot corresponding to our test image set, comparing the distributions of these scores (M5-M7) given to histochemical and virtually stained images. For metrics M6 and M7, the expert score distributions for the virtually stained images surpassed those of the histochemically stained images, with mean improvements of 0.17 (5.67%) and 0.24 (8.33%) for M6 and M7, respectively. For M5, however, the mean difference in performance dropped to an insignificant level of 0.0066 (0.22%) in favor of the histochemically stained images. Figure 4b further depicts the mean and the standard

deviation values across all the samples for each pathologist (also see Supplementary Table 1).

Note that because the histochemically stained and virtually stained images for the same tissue FOVs were both included in the expert evaluation process, we can also compare the pathologists' scores for individual image FOVs. For this image pair-based analysis, we first compared the average scores of the 3 pathologists on the same image FOVs, generated for the histochemically stained and virtually stained images. The results of this paired analysis are displayed in Fig. 4c, where each column contains three metrics for a given FOV. A yellow-colored entry represents a higher score given to the virtually stained image output for that tissue FOV, while a green color for an entry represents that a lower score is given to the virtually stained image FOV compared to the histochemically stained counterpart; finally, a gray color represents equal scores. These analyses reported in Fig. 4c reveal that the virtually stained label-free images were blindly given higher or equal performance scores by the expert panel in 66%, 91%, and 89% of the tissue FOVs for metrics M5, M6 and M7, respectively. A more detailed comparison of each pathologist's individual scores is also reported in Supplementary Fig. 6. Additional image FOVs can be found in Supplementary Fig. 7, and two cases where the virtually stained images received lower and equal scores are reported in Supplementary Fig. 8. These quantitative comparisons further demonstrate the success of our virtual birefringence imaging and label-free tissue staining method using deep learning.

### Virtual staining model quantitative evaluation

To further validate our results, we employed several quantitative metrics (see the Methods section and Supplementary Fig. 9) to assess the level of agreement between virtually and histochemically stained images. We initially focused on birefringence images due to their clinical significance in amyloidosis detection. Our analysis comprised two aspects: the area of the apple-green regions and the color distribution of the images; see Fig. 5. The table in Fig. 5a presents the image comparison metrics that we used: mean absolute error (MAE), multiscale structural similarity index metric (MS-SSIM), peak signal-to-noise ratio (PSNR), and Fréchet inception distance (FID). The metric values (low MAE and FID; high MS-SSIM and PSNR) and low standard deviations indicate a strong agreement between our virtually stained images and the corresponding histochemical ground truth. We also segmented the apple-green birefringent regions and measured the accuracy of our predictions using down-sampled intersection-over-union (D-IoU); see the Methods section for details. Two example FOVs with their segmented masks and the corresponding D-IoU values are shown in Fig. 5b. Figure 5c also displays the D-IoU distribution of all the samples, highlighting the high concordance between the output images and the ground truth, where the regions containing amyloid deposits in the output images overlap significantly with those in the target images.

Next, we assessed the color distribution of (1) the segmented apple-green areas and (2) the entire images. We converted the virtual and histochemical images of two FOVs into YCbCr channels and plotted the histograms of each channel where the luminance information is stored as a single component (Y), and the chrominance information is stored as two color-difference components (Cb and Cr). The results, shown in Fig. 5d, demonstrate a very good agreement in the color distribution, validating the color accuracy of our model inference. To further confirm the color accuracy, we calculated the pixel value distribution across the entire testing dataset and plotted the results in Supplementary Fig. 10, which revealed a strong concordance across all the channels, as desired.

For the brightfield Congo red images, we applied similar quantitative metrics (MAE, MS-SSIM, PSNR, and FID), as shown in Supplementary Fig. 11a; we also included two additional metrics commonly used for brightfield images: the total count and the average size of nuclei in an image per FOV. The results are displayed in the scatter plots shown in Supplementary Fig. 11b. These results further affirm the concordance between our virtual staining results and the histochemically stained ground truth brightfield images, demonstrating the effectiveness of our method.

### Transfer learning for noise resilience

To demonstrate our model's ability to handle variations in image signal-to-noise (SNR), we utilized transfer learning to bring resilience against noise. The results of this analysis are summarized in Supplementary Fig. 12 where we digitally added Gaussian noise to simulate autofluorescence images captured with lower SNR image sensors. Direct inference using those noisy autofluorescence images as input to a model trained on original images (before the addition of Gaussian noise) generated hallucinated output images that are not acceptable. However, after applying transfer learning to the original model using noise-added images, the model quickly learned to adapt to the noisy inputs and inferred stained images that closely match the histochemical ground truth images (see Supplementary Fig. 12). This showcases that our virtual staining model can be rapidly fine-tuned through transfer learning to effectively adapt to new configurations and maintain high fidelity.

## Discussion

In this manuscript, we introduced a deep learning-based approach that combines virtual birefringence imaging and virtual Congo red staining, enabling label-free imaging and detection of amyloid deposits. Specifically, we leveraged tissue autofluorescence texture at the sub-micron scale to generate virtual images that mimic the Congo red-stained tissue brightfield and polarization images, facilitating accurate identification of amyloid deposits within label-free tissue.

In recent years, there have been accelerated efforts to overcome some of the challenges seen in traditional glass slide-based pathology. These have led to the development and adoption of digital imaging systems and WSI scanners that have helped with the transition of pathology into the digital era. While WSI scanners have demonstrated their capabilities to digitize various histochemical and immunohistochemical stained slides, automatically digitizing slides under polarized light remains technically challenging in histology labs due to delicate polarization components and varying illumination conditions. When visualizing a tissue slide under polarized light, constant adjustments of the tissue orientation and polarizer are required to ensure accurate and robust detection of birefringence patterns. In fact, none of the commercially available, clinically approved WSI scanners in the digital pathology field can automatically digitize birefringence images of tissue samples. As a result, pathologists continue to rely on manually operated light microscopes and standard polarizers for amyloid deposit inspection/detection, which is also at the heart of diagnostician-induced errors. Moreover, the quality of the light microscope and polarizers significantly influences the detection ability of amyloidosis by the practicing pathologist. In instances with low amounts of amyloid deposits within the tissue, the microscope's ability to visualize the slide with high contrast and resolution can make the difference between a correct and false diagnosis. By training a virtual staining neural network to transform label-free tissue autofluorescence images into birefringence and brightfield microscopy images that are equivalent to the Congo red stained images of the same tissue, as it would normally appear after chemical staining, we offer a deep learning-based solution to both the challenging nature of the histochemical Congo red staining process as well as the digitization of the birefringence images of stained tissue sections, which currently does not exist in clinical WSI scanners. Our virtual polarization imaging and tissue staining method ensures consistent and reproducible imaging of amyloid deposits within label-free tissue, eliminating the manual processes needed in both the chemical staining

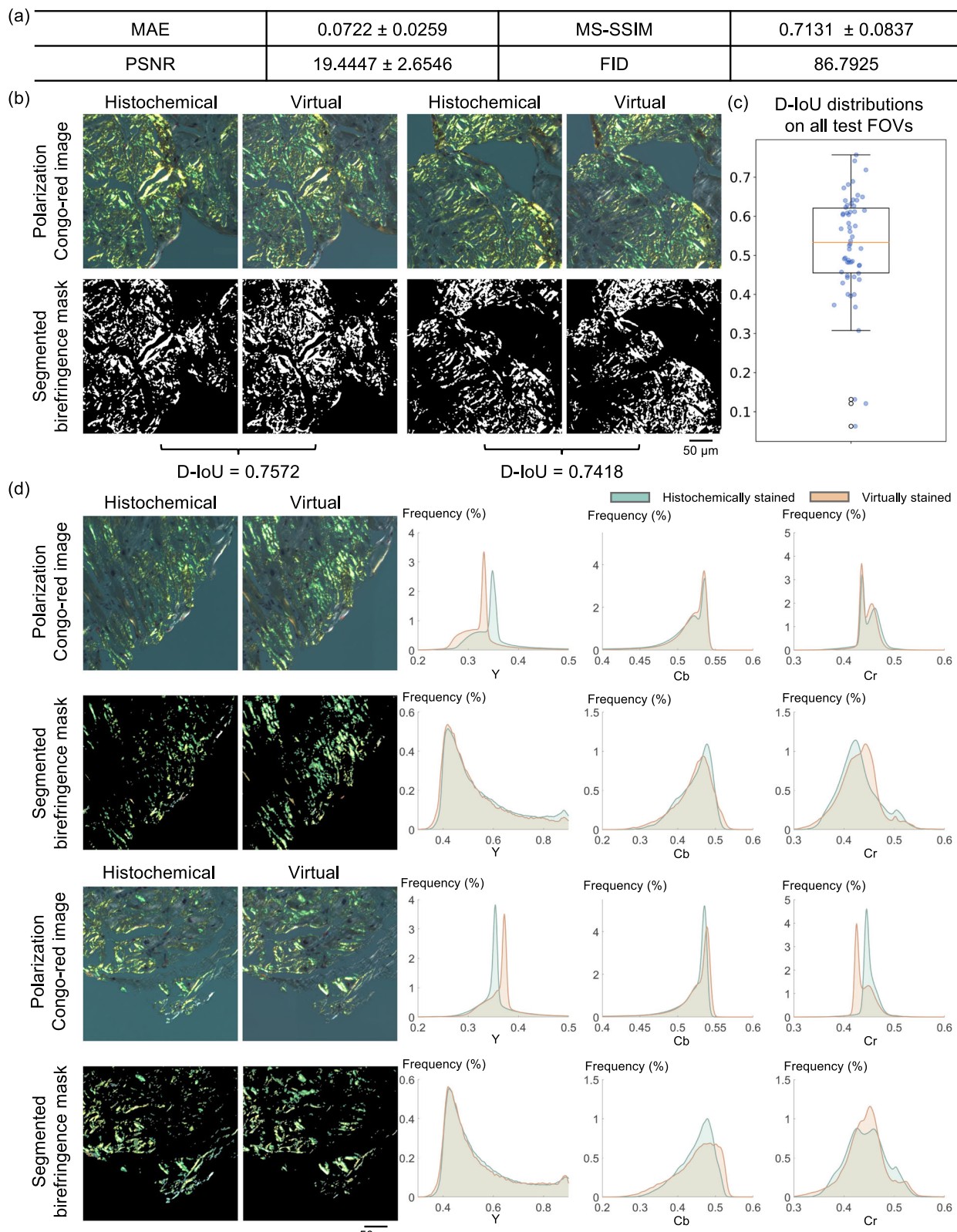

of tissue and the constant polarizer and tissue adjustments routinely performed by diagnosticians when using a polarized light microscope. Bypassing such manually operated polarization microscopes, our method can be readily adopted on standard pathology WSI scanners approved for digital pathology by simply using standard filter sets commercially available for capturing autofluorescence images of tissue slices. This also eliminates the need for microscopy hardware

changes or specialized optical parts in digital pathology scanners that are already deployed. In addition to cross-polarized imaging, our system can also enable pathologists to virtually rotate the polarization filters by a small angle. This introduces an additional capability to virtual Congo red staining and can be used to further validate the presence of amyloid deposits, which should change from an apple-green color to a yellow-like color with -10 degrees rotation of the

**Fig. 5 | Quantitative evaluation results for comparing the histochemically and virtually stained polarization Congo red images. a** The table lists mean absolute error (MAE), multiscale structural similarity index metric (MS-SSIM), peak signal-to-noise ratio (PSNR), and Fréchet inception distance (FID) between the histochemically and virtually stained polarization Congo red images. **b** Two examples of down-sampled intersection-over-union (D-IoU) measurements between the segmented apple-green birefringence masks from histochemically and virtually stained polarization Congo red images. **c** The D-IoU distributions across all 56 testing pairs (histochemically and virtually stained fields-of-view (FOVs)). For the box plot, the median

represents the central value. The box marks the interquartile range, with the lower and upper quartiles defining its boundaries. Whiskers extend from the box to encompass data points within 1.5 times the interquartile range; values beyond this range are considered outliers and plotted individually. All data points are overlaid on the plot with x-axis jitter to enhance visualization. **d** Two examples of color histograms in YCbCr color space for the whole FOV and the birefringence regions only. The blue curves represent the distributions of histochemically stained images, while the red ones present the virtually stained images. Source data of the quantitative evaluation metric values for all 56 testing FOV pairs are provided as a Source Data file.

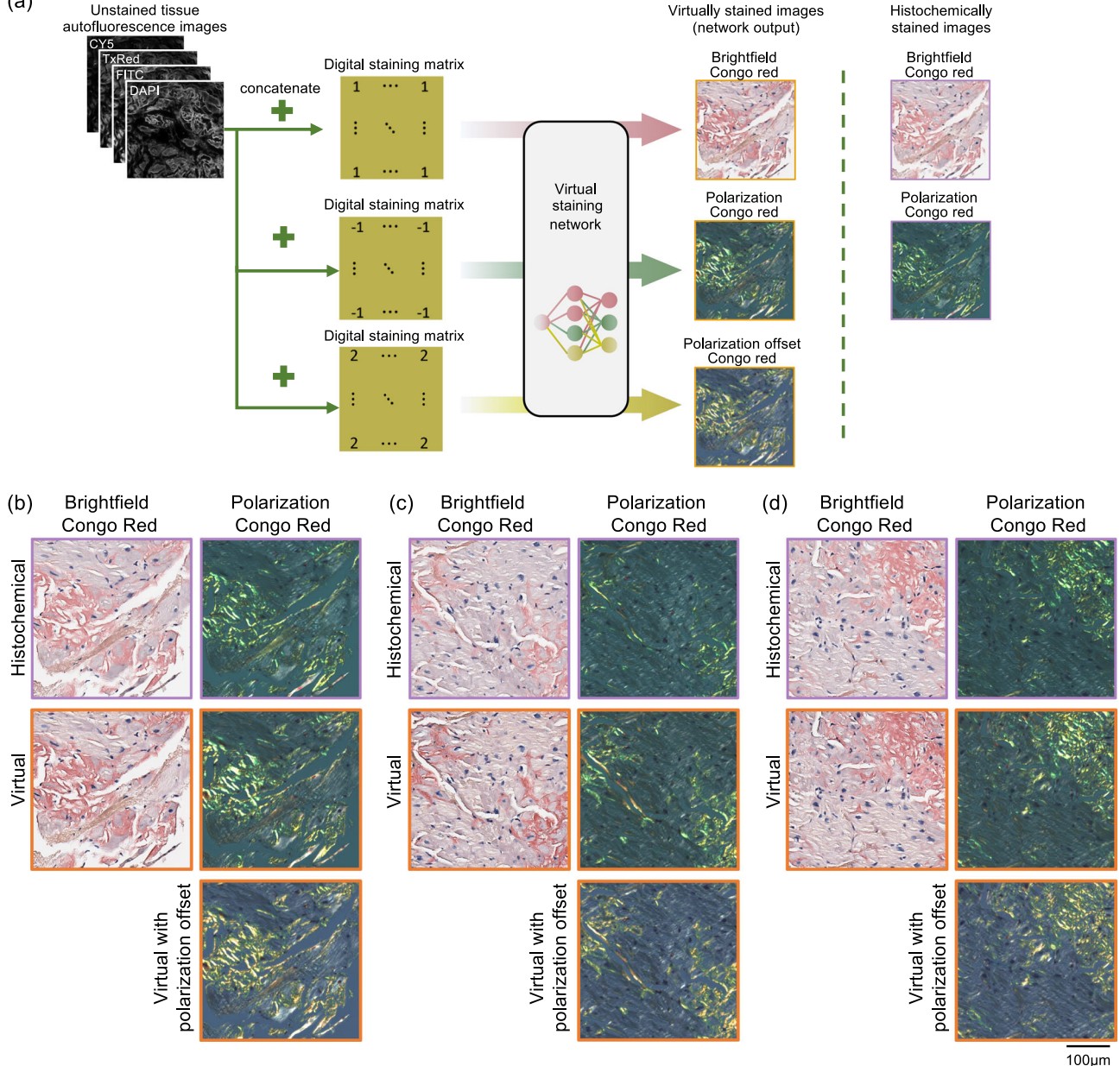

**Fig. 6 | Virtual birefringence imaging with a shifted angle. a** The virtual staining neural network is trained to simultaneously generate brightfield, cross-polarized birefringence, and birefringence images with a shifted angle using three digital staining matrices having pixel values of 1, −1 and 2, respectively. **b–d** Three separate fields-of-view are shown in three panels. The top row displays histochemical

brightfield and polarization Congo red images. The second row presents virtually stained brightfield and cross-polarized birefringence images. The bottom row shows virtual polarization images with a shifted angle that features yellow-colored amyloid deposits.

polarization filter[39]. For its proof-of-concept, we demonstrated this capability by adding another channel in the DSM, where a pixel value of 2 in the staining matrix represents the angle-shifted birefringence channel, as shown in Fig. 6a. This new neural network, with 3 different

virtually stained output images in its inference, was then trained using digitally emulated images where the color of the amyloid deposits changed from apple-green to yellow (see the Methods section), as expected from a slight shift of the polarizer angle. The blind testing

results of this new virtual staining network, shown in Fig. 6b, indicate that it successfully generated birefringence output images where the amyloid deposits appear in yellow, in addition to generating the other two channels, i.e., the cross-polarized and brightfield images. These results serve as a proof-of-concept demonstration of the DSM's multiplexing capability and its potential for additional output channels to further assist diagnosis by virtually transforming the polarization filter into different states, as desired.

Quantitative and comparative analyses of three board-certified pathologists revealed that the virtually stained slides, with their bire-fringence and brightfield image channels, were on par with chemically stained Congo red slides. Since our label-free approach is not critically dependent on manual labor in its staining and imaging processes, it is particularly beneficial in cases with low amounts of amyloid deposits, which can be easily missed when a diagnostician examines the slide without a high-quality polarizing microscope. In such cases, our method may increase the diagnostic yield and decrease false negative rates. An interesting future direction can be automated label-free detection of regions of amyloid deposits to highlight the problematic areas, which may further assist pathologists with attention heatmaps to accelerate diagnosis speed and reduce false negative rates.

We also conducted an ablation study (see Supplementary Fig. 13) to show that with the decreasing number of input autofluorescence channels, the output image quality of the virtual staining network decreases. This indicated that four input autofluorescence channels (DAPI, FITC, TxRed and Cy5) are indeed essential to acquire the necessary information for successfully generating the virtually stained images of amyloid deposits.

The presented label-free approach has several inherent advantages compared to traditional histochemical Congo red staining. It minimizes staining artifacts compared to chemical staining techniques; it substantially reduces reliance on manual labor, and the virtual staining process diminishes the utilization of hazardous chemicals during slide preparation. It is important to emphasize that all the training processes described for the virtual staining algorithm, including the use of different modalities and processing pipelines, are a one-time effort. Once the model is trained, the blind inference process (virtual staining with brightfield and birefringence channels) of a new, unknown sample will only require seconds per FOV (e.g., the inference time for a FOV size of 2048 × 2048 pixels is <2 s); therefore, it can transform a label-free whole slide within a few minutes using a state-of-the-art GPU.

In addition to the technical challenges in amyloidosis detection, the spotty nature of the disease and variations in the density of amyloid deposits increase the odds of false diagnoses. Thick tissue sections (e.g., 8–10 μm) are recommended for accurate Congo red birefrin-gence visualization, as they provide more intense staining and allow for the identification of smaller amyloid deposits compared to the commonly used 4 μm sections in pathology. Nonetheless, thicker sections can negatively affect the amount of residual tissue and may have a detrimental impact on the depletion of small biopsy blocks, such as those used in cardiac biopsies. Insufficient tissue is often a limiting factor in performing additional stains or molecular studies. We believe that our method, relying on tissue autofluorescence rather than imaging under polarized light, would be less sensitive to tissue thickness and potentially can assist clinicians in reaching the same diagnostic conclusions while sparing tissue.

The virtual staining network's performance was superior in the polarization Congo red channel and comparable in the brightfield channel, when benchmarked against the histochemical ground truth. This change in behavior might be due to the trade-off introduced by the DSM: compared to training two distinct neural networks for brightfield Congo red and birefringence image generation, where an optimal model for each modality could be created, employing a DSM within a single neural network may result in a model that is optimal for

one imaging modality but slightly suboptimal for the other. On the other hand, DSM inference provides better structural alignment between the output channels (as quantified by M7), whereas two separate network models might produce divergent artifacts and inconsistencies, potentially confusing pathologists during diagnosis; that is why, we utilized the presented architecture of the DSM-based multi-modal inference to bring consistency between virtually generated polarization and brightfield channels.

A limiting factor in our study was the scarcity and limited size of cardiac biopsy samples that were available to us. We trained and tested our virtual Congo red staining on tissue slides obtained from a single pathology laboratory, which may not exhaustively represent all clinically significant features observable in cardiac amyloidosis samples. For example, other tissue components like collagen can exhibit some birefringence that differs in appearance from amyloid fibrils. It is important to note that Congo red staining highlights amyloid deposits, and stains collagen/elastic fibers with far less affinity[40]. Therefore, the typical apple-green birefringence signatures distinguish amyloid deposits from other fibrils and from the white birefringence of fibrin or collagen[41]. Moreover, since our virtual staining approach does not rely on birefringence images at its input channel and only uses auto-fluorescence images of label-free tissue, our approach should, in principle, be immune to various forms of non-specific tissue birefrin-gence as long as their microscopic spatial features at multiple auto-fluorescence image channels do not substantially overlap with the spectral and spatial features of label-free amyloid autofluorescence. Thus, the fact that our input images utilize four different channels of autofluorescence (i.e., DAPI, FITC, TxRed and Cy5) helps with the specificity of our virtual staining approach for amyloid deposits. To further shed light on this, future work will involve validating our model's capability to accurately differentiate mimics of amyloidosis from amyloid deposits, based on their label-free autofluorescence images. We also plan to expand our evaluations to other tissue types, such as kidney, liver, and spleen, using tissue slides from additional histology laboratories, which will further enhance the generalization and inference performance of our model.

In conclusion, we reported the demonstration of virtual birefrin-gence imaging and virtual tissue staining that effectively transforms label-free cardiac tissue slides into images that match their Congo red-stained histochemical counterparts, viewable in both brightfield and polarization channels. Our method, as a fully digital process, can generate virtually stained WSIs in minutes with high repeatability, eliminating the variations and limitations associated with manual tissue staining methods and the use of polarization microscope components. By reducing the turnaround time, manual labor, and reliance on hazardous chemicals, this method can potentially transform the traditional workflow for the diagnosis of amyloidosis and set the stage for a large-scale, multi-center trial to further validate the clinical utility of our results.

## Methods

### Sample preparation and data acquisition

Unlabeled heart tissue sections were obtained from USC Keck School of Medicine under Institutional Review Board (IRB) #HS-20-00151 with ethical approval granted. Following the clinical standard for amyloid inspection, 8 μm sections were cut from archived cardiac tissue blocks that tested positive for Congo red. Samples with less than 5 mm² tissue area or less than 5% amyloid-involved tissue, determined by the original pathology report, were excluded. After autofluorescence scanning, the standard Congo red staining was performed.

Autofluorescence images of the aforementioned label-free car-diac tissue samples were taken using a conventional scanning fluorescence microscope (IX-83, Olympus) equipped with a ×40/0.95NA objective lens (UPLSAPO, Olympus). These images were captured at four distinct excitation and emission wavelengths, each using a

fluorescent filter set in a filter cube: DAPI (Semrock DAPI-5060C-OFX, EX 377/50 nm, EM 447/60 nm), FITC (Semrock FITC-2024B-OFX, EX 485/20 nm, EM 522/24 nm), TxRed (Semrock TXRED-4040C-OFX, EX 562/40 nm, EM 624/40 nm), and Cy5 (Semrock CY5-4040C-OFX, EX 628/40 nm, EM 692/40 nm). Note that the selection of the filters was primarily empirical, based on the availability of commonly used fluorescence filter sets, which spanned the visible spectrum while having spectral selectivity to capture distinct autofluorescence bands effectively[42]. The autofluorescence images were recorded using a scientific complementary metal-oxide-semiconductor (sCMOS) image sensor (ORCA-flash4.0 V2, Hamamatsu Photonics) using exposure times of 150 ms, 500 ms, 500 ms, and 1000 ms for the DAPI, FITC, TxRed, and Cy5 filters, respectively. The exposure time for each channel was selected based on the intensity levels observed, which helped us to use a substantial portion of the dynamic range of the image sensor. µManager (version 1.4) software[43], designed for microscope management, was used for the automated image capture process. Autofocus[44] is applied on the first autofluorescence channel (DAPI) for each FOV. Following the completion of the standard Congo red staining, high-resolution brightfield WSIs were obtained using a scanning microscope (AxioScan Z1, Zeiss) with a ×20/0.8NA objective lens (Plan-Apo) at the Translational Pathology Core Laboratory (TPCL) at UCLA. Polarized images of Congo red-stained slides were captured using a modified conventional brightfield microscope (IX-83, Olympus) with a halogen lamp used as the illumination source without blue filters. The microscope was equipped with a linear polarizer (U-POT, Olympus) and an analyzer with an adjustable wave plate (U-GAN, Olympus) with a ×20/0.75NA objective lens (UPlanSApo). The linear polarizer was placed on the condenser adapter (between the light source and the sample slide) and fixed in a customized 3D printed holder (Ultimaker S3, PETG black) to ensure polarization orientation during the scanning process. The analyzer was inserted into the slider-compatible revolving nosepiece, between the sample slide and the image sensor. The orientations of all polarization components were adjusted by a board-certified pathologist for optimal image quality.

### Image preprocessing and registration

The registration process is essential to successfully train an image-to-image translation network. An alternative is to apply CycleGAN-like architectures[33,45] which do not require paired samples for training. However, such unpaired image-based approaches result in inferior image quality with potential hallucinations compared to training with precisely registered image pairs[18,21,38]. In this work, we registered both the brightfield and birefringence images individually to match the autofluorescence images of the same samples using a two-step registration process. First, we stitched all the images into WSIs for all three imaging modalities and globally registered them by detecting and matching speeded-up robust features (SURF) on downsampled WSIs[46,47]. Then, we estimated the spatial transformations (projective) using the detected features with an M-estimator sample consensus algorithm[48] and accordingly warped the brightfield and birefringence WSIs. Following this, the coarsely matched autofluorescence, brightfield and birefringence WSIs were divided into bundles of image tiles, each consisting of 2048 × 2048 pixels. Using these image tiles, we further improved the accuracy of our image registration to address optical aberrations among different imaging systems and morphological changes that occurred in the histochemical staining process with a correlation-based elastic registration algorithm[18,49]. During this elastic registration process, a registration neural network model was trained to align the style of the autofluorescence images with the styles of the brightfield and birefringence images. The registration model used for this purpose shared the same architecture and training strategy as our virtual staining network (detailed in the next section), however, without a registration submodule and was only used for the data preparation stage. After the image style transformation using the

registration model, the pyramid elastic image registration algorithm was applied. This process involved hierarchically matching the local features of the sub-image blocks of different resolutions and calculating transformation maps. These transformation maps were then used to correct the local distortions in the brightfield and birefringence images, resulting in a better match with their autofluorescence counterparts. This training and registration process was repeated for both brightfield and birefringence images until precise pixel-level registration was achieved. The registration steps were implemented using MATLAB (MathWorks), Fiji[50] and PyTorch[51]. To simulate the polarization filter rotation and generate images with yellow-colored amyloid deposits, we first extracted the apple-green part of an image using a segmentation algorithm (see the "Quantitative evaluation metrics for polarization Congo red virtual staining" section). Then, we multiplied the hue channel by a fixed parameter (-0.6) to transform the color of the amyloid deposits from apple-green to yellow. Meanwhile, we also multiplied the hue channel of the background regions by -1.1 to simulate the background color change to dark blue. These parameters were tuned and approved by a pathologist.

### Quantitative evaluation metrics for brightfield Congo red virtual staining

To quantitatively evaluate the performance of brightfield Congo red virtual staining, we organized 56 FOVs of virtually generated brightfield Congo red images together with their corresponding histochemically stained images for paired image comparison. Similar to refs. 23,52. we first used standard metrics of MAE, MS-SSIM, PSNR, and FID, as well as some other customized metrics for brightfield images, including the number of nuclei per FOV, and average area of nuclei per FOV, as shown in Supplementary Fig. 9a. MAE was defined as,

$$MAE = \frac{1}{MN} \sum_m \sum_n |A(m,n) - B(m,n)| \tag{1}$$

where $A, B$ represent histochemically and virtually stained brightfield Congo red images, respectively, $m$ and $n$ are the pixel indices, and $M \times N$ denotes the total number of pixels in each image.

The MS-SSIM[53] evaluated the SSIM between two images at different spatial levels, where the number of scales was set to 6, and the weights applied to each scale were set to [0.05, 0.05, 0.1, 0.15, 0.2, 0.45].

The PSNR was calculated using,

$$PNSR = 10 \log_{10} \left( \frac{\max(A)^2}{MSE} \right) \tag{2}$$

where $A$ presents the ground truth image (histochemically stained), and MSE was defined as,

$$MSE = \frac{1}{MN} \sum_m \sum_n [A(m,n) - B(m,n)]^2 \tag{3}$$

The FID was calculated according to the definition reported in ref. 54. using the default feature number.

As for the quantifications of nuclei properties, we adopted a stain deconvolution method[55] to first separate the channel corresponding to nuclei staining. Then, Otsu's thresholding[56] and a series of morphological operations, such as image dilation and erosion, were applied to the nuclei channel to obtain the segmented binary nuclei mask. The number of nuclei per FOV was defined as the number of connected components in the binary nuclei mask, and the average area of nuclei per FOV corresponds to the average area of connected components across the whole binary nuclei mask with a unit of pixel[2].

## Quantitative evaluation metrics for polarization Congo red virtual staining

To perform the quantitative evaluation of polarization Congo red virtual staining, a paired comparison was conducted on 56 paired histochemically and virtually stained images with the same FOVs used in the brightfield image evaluation. We also used the metrics of MAE, MS-SSIM, PSNR, and FID as defined earlier. There were also some differences: (1) for MS-SSIM, the weights applied on each scale are [0.45, 0.2, 0.15, 0.1, 0.05, 0.05]; (2) for the calculation of FID, to adjust the image brightness, we transferred both the histochemically and virtually stained polarization images into YCbCr color space, multiplied the Y-channel values by 1.5, and then transferred the image back to the ordinary RGB color space. Additionally, we developed a segmentation algorithm to isolate the apple-green birefringence regions in polarization Congo red images by empirically setting thresholds on the HSV channels. The Hue channel in the HSV space enabled the direct selection of the green color range for amyloid deposits. Thresholds were determined from sample FOVs and then validated and finetuned by a board-certified pathologist to ensure accurate amyloid segmentation. Subsequently, we applied several morphological operations, such as image opening and image closing. As shown in Supplementary Fig. 9b, we used D-IoU metric to measure the similarity between the segmented birefringence masks of the histochemically and virtually stained polarization Congo red images, defined as follows,

$$D - IoU = \frac{\sum_m \sum_n [A_{32}(m,n)^* B_{32}(m,n)]}{\sum_m \sum_n [\min\{(A_{32}(m,n) + (B_{32}(m,n)), 1\}]} \quad (4)$$

where $A_{32}$ and $B_{32}$ correspond to the 32× down-sampled (bilinear) version of the segmented birefringence masks for the histochemically and virtually stained polarization Congo red images, respectively, and $m$ and $n$ denote the pixel indices. Before computing D-IoU, $A_{32}$ and $B_{32}$ were binarized with a small threshold for logical operations. Compared to the traditional per-pixel definition of IoU, this down-sampled D-IoU is more suitable to measure the similarity between binary masks and is resilient to small pixel misalignments[57]. Finally, for the whole FOV and the apple-green birefringence regions only, we drew the color distributions using probability density functions fitted from histograms separately calculated in Y, Cb, and Cr channels to compare the color similarity between the histochemically and virtually stained polarization Congo red images.

## Network architecture and training strategy

We utilized a conditioned GAN architecture[36,58] to learn the transformation from the 4-channel label-free autofluorescence images (DAPI, FITC, TxRed, and Cy5) into the corresponding microscopic images of Congo red stained tissue under brightfield microscopy and polarized light microscopy. As shown in Fig. 1b, this conditioned GAN employed a DSM, digitally concatenated with the autofluorescence input images, with all the elements of the matrix set to "1" corresponding to *brightfield* Congo red images, while "−1" refers to *birefringence* images. The conditioning of this DSM can be represented as:

$$\widetilde{c} = [c_1] \, or \, [c_{-1}] \quad (5)$$

where $\widetilde{c}$ is concatenated to the input as an additional channel. $[c_1]$ and $[c_{-1}]$ denote matrices with all the elements equal to 1 and -1, respectively. Note that for the additional 3rd channel required for yellow-colored amyloid deposits (see Fig. 6a), Eq. 5 is expanded to $\widetilde{c} = [c_{-1}] \, or \, [c_1] \, or \, [c_2]$.

The GAN framework consists of two deep neural networks, a generator and a discriminator. During the training, the generator is tasked with learning a statistical transformation to create virtually stained images. Concurrently, the discriminator learns to distinguish the generated virtually stained images and their histochemically stained counterparts. This competitive training paradigm fosters the simultaneous enhancement of both neural networks. In our training, the generator (G) and the discriminator networks (D) were optimized to minimize the following loss functions:

$$L_{generator} = \alpha L_1(I_{target}, G(I_{input}, \widetilde{c}) \circ R(G(I_{input}, \widetilde{c}), I_{target})) \\ + \beta BCE(D(G(I_{input}, \widetilde{c}), \widetilde{c}), 1) + \gamma TV(G(I_{input}, \widetilde{c})) \quad (6)$$

$$L_{discriminator} = BCE(D(G(I_{input}, \widetilde{c})), 0) + BCE(D(I_{target}), 1) \quad (7)$$

where G(·) and D(·) refer to the outputs of the generator and discriminator networks, respectively; $I_{target}$ denotes the ground truth image; and $I_{input}$ denotes the input label-free autofluorescence images. The binary cross-entropy (BCE) loss is defined as:

$$BCE(p(a), p(b)) = -[p(b)\ln(p(a)) + (1 - p(b))\ln(1 - p(a))] \quad (8)$$

where $p(a)$ represents the discriminator prediction and $p(b)$ represents the actual label (0 or 1). The total variation (TV) loss acted as a regularizer term and can be defined as:

$$TV(I) = \sum_p \sum_q |I_{p+1,q} - I_{p,q}| + |I_{p,q+1} - I_{p,q}| \quad (9)$$

where $p$ and $q$ are the pixel indices of the image $I$. The coefficients $(\alpha, \beta, \gamma)$ in Eq. 6 were empirically set as $(10, 1, 0.0001)$ based on the validation image set and were fixed for the finalized model evaluation on test samples. In Eq. 6, we also incorporated an additional registration module $R$ (as shown in Fig. 7a) into our virtual staining network to remove the residual alignment errors between the generated images and their corresponding ground truth images. The registration module was fed with a pair of images: a 'fixed' image, which was the ground truth, and a 'moving' image, the virtually generated one. This registration module outputs a *transformation/displacement matrix* detailing pixel displacements needed to align the virtually stained image with the ground truth[59]. Using the displacement matrix, a grid flow matrix containing new locations of each pixel was generated and applied to the virtually stained images via a resampling operation (denoted as ∘), which was implemented using PyTorch. During the training, the loss function of the registration module was defined as:

$$L_{registration} = \lambda L_1(I_{target}, G(I_{input}, \widetilde{c}) \circ R(G(I_{input}, \widetilde{c}), I_{target})) \\ + \mu SMTH(R(G(I_{input}, \widetilde{c}), I_{target})) \quad (10)$$

where the smooth loss[60] (SMTH) is defined as:

$$SMTH(T) = \frac{1}{XY} \sum_x \sum_y |T_{x+1,y} - T_{x,y}|^2 + |T_{x,y+1} - T_{x,y}|^2 \quad (11)$$

where $T$ is the displacement matrix. $X \times Y$ denotes the total number of elements in the matrix; $x$ and $y$ are the element indices. The coefficients $(\lambda, \mu)$ in Eq. 6 were empirically set as $(20, 10)$.

Note that this registration module $R$ is not the same as the registration models mentioned in the previous section: a registration model learns the style-transfer to help with the elastic registration and is solely used in the data preparation stage. Unlike other shift-invariant losses used in earlier work[61], the registration module in our work was trained simultaneously with the virtual staining model to learn the misalignments between the target and output images from the virtual staining network. These misalignments include not only static optical aberrations but also complex tissue distortions introduced during the

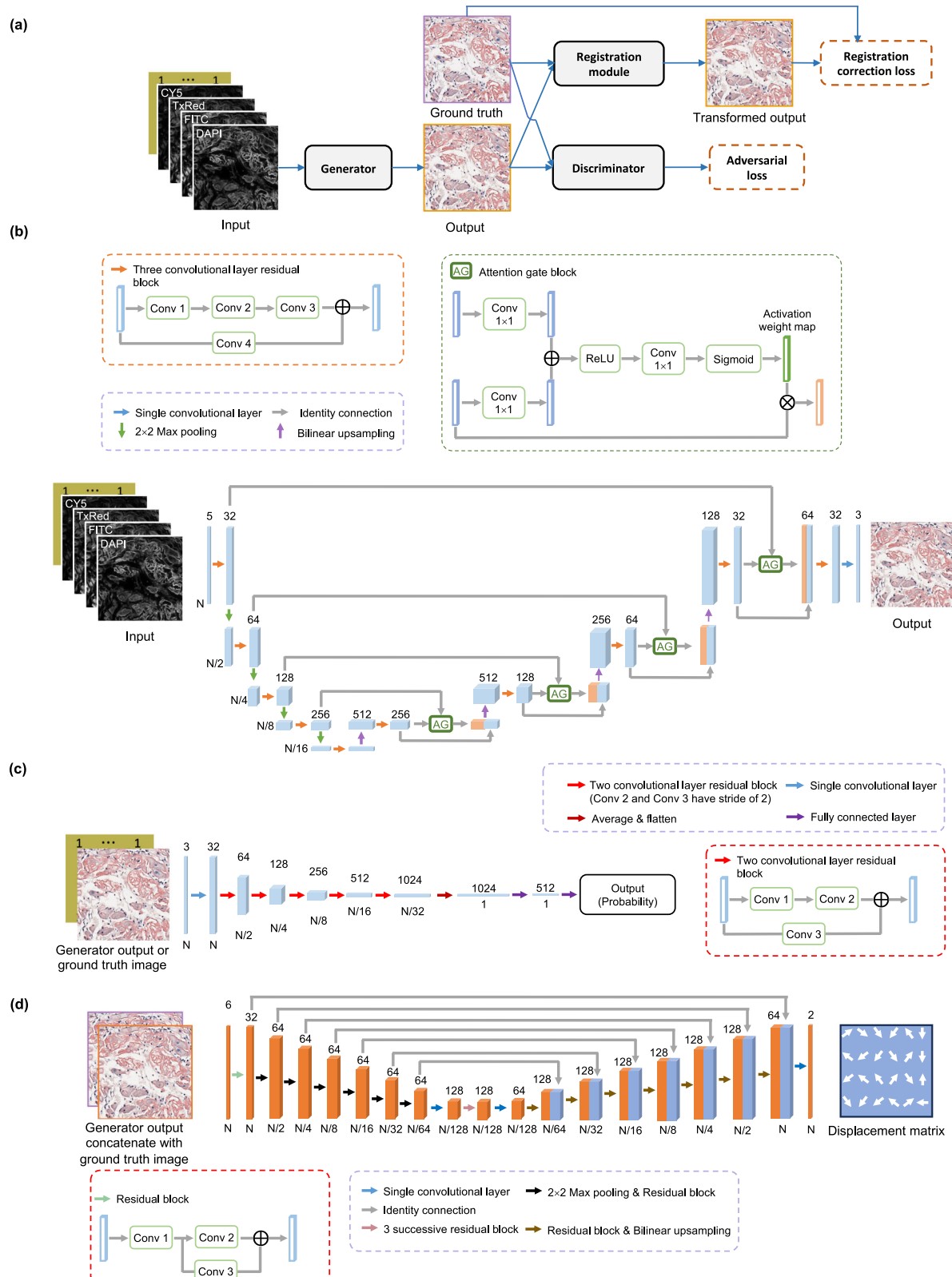

**Fig. 7 | Network architecture for virtual birefringence imaging and virtual staining of amyloid deposits in label-free tissue using autofluorescence microscopy and deep learning. a** The overview of the generator, discriminator and registration module. **b** Detailed architecture and building blocks of the generator. **c** Detailed architecture and building blocks of the discriminator. **d** The detailed architecture of the registration module.

histochemical staining process, which can vary from one tissue FOV to another. These distortions are difficult to characterize with a simple loss function or data augmentation techniques but can be effectively learned with an optimizable network module.

Both the discriminator and registration modules were only used in training; during the testing phase, only the generator network was used with a simple forward process to infer the virtually stained images.

To avoid possible exploding gradients, we used the smooth $L_1$ loss[62] represented as:

$$L_1(A, B) = \frac{1}{MN} \left( \sum_{|A(m,n)-B(m,n)|<\varphi} 0.5 \frac{(A(m,n)-B(m,n))^2}{\varphi} + \sum_{|A(m,n)-B(m,n)|\geq\varphi} |A(m,n)-B(m,n)| - 0.5\varphi \right) \quad (12)$$

where $A, B$ represent the images in the comparison, $m$ and $n$ are the pixel indices, and $M \times N$ denotes the total number of pixels in each image. $\varphi$ was empirically set to 1.

Figure 7b depicts the generator network modeled on the attention U-Net architecture[63]. This network is composed of a series of four downsampling blocks and four upsampling blocks. Each downsampling block comprises a three-convolution-layer residual block, followed by a leaky rectified linear unit (Leaky ReLU[64]) with a slope of 0.1, and a 2 × 2 max pooling layer with a stride size of 2, which downsample the feature maps and double the number of channels. The three-convolution-layer residual block[65] is formed by three consecutive convolution layers and a convolutional residual path building a bridge between the input and output tensors of the residual block.

The input for each upsampling block is a fused tensor, concatenating the output from the preceding block with the corresponding feature maps at the matched level of the downsampling path passing through the attention gate connections. The attention gate passes a tensor through three convolution layers and a sigmoid operation, generating an activation weight map applied to the tensor to strengthen salient features[63]. The upsampling blocks bilinearly resize (2×) the concatenated tensors and then use the three-convolution-layer residual block to reduce the number of channels by a factor of four. The final upsampling block was followed by a three-convolutional layer residual block together with another single convolution layer, reducing the number of channels to 3, matching the ground truth images.

The discriminator network, as depicted in Fig. 7c, processes the inputs that are either the virtually stained images generated by the network or the actual ground truth images. It begins by transforming the input image into a 64-channel tensor via a single convolutional layer, followed by activation through a Leaky ReLU. Then, the resulting tensor passed through five successive two-convolutional-layer residual blocks, each of which set the stride size of the second convolutional layer as 2 for 2× downsampling and doubling the number of channels. These blocks were followed by a global pooling layer and two dense layers to obtain the output probability of the input being the ground truth (histochemically stained) image.

For the registration module[37], as depicted in Fig. 7d, we adopted a U-net architecture akin to that of the generator, albeit with several key modifications. The network comprises seven pairs of downsampling and corresponding upsampling blocks, each integrated with a residual block. Following the final downsampling stage, a convolution layer is employed to double the feature channels, succeeded by a series of three consecutive residual blocks. Subsequently, another convolution layer halved the channels, transitioning the processed features to the upsampling block sequence. The output layer of the network utilizes a single convolution layer to condense the number of channels to 2, corresponding to the two normal components (x and y) of the displacement matrix.

The training dataset contained 386 image patches, each with dimensions of 2048 × 2048 pixels, extracted from 8 distinct patients. During the training, the networks received image patches sized 256 × 256 pixels, randomly cropped from the larger 2048 × 2048 patches in the dataset. The generator, discriminator and registration modules were all optimized using the Adam optimizers[66], starting with learning rates of $2 \times 10^{-5}$, $2 \times 10^{-6}$ and $2 \times 10^{-6}$, respectively. A batch size of 32 was maintained throughout the training phase. The generator/discriminator/registration module update frequency was set to 4:1:1. The network converged after ~48 hours of training. The training and testing are done on a standard workstation with GeForce RTX 3090 Ti graphics processing units (GPU) in workstations with 256GB of random-access memory (RAM) and Intel Core i9 central processing unit (CPU). The virtual staining network was implemented using Python version 3.12.0 and PyTorch[51] version 1.9.0 with CUDA toolkit version 11.8.

## Pathologists' blind evaluations

Three board-certified pathologists were included to blindly evaluate the image quality of histochemically stained and virtually stained cardiac tissue sections. The blind evaluations were conducted in two ways: (1) Brightfield Congo red stain image quality on small image patches with M1–M4, which are standard quantification metrics for histology images; and (2) Large patch bundled images with both brightfield and birefringence images of the same FOV. In anatomic pathology, the evaluation of Congo red-stained slides under polarized light microscopy yields a binary result—positive or negative for amyloidosis. To enhance the quantification of our virtual polarization model, we incorporated additional metrics: amyloid quantification (M5), the quality of birefringence appearance (M6), and the consistency between brightfield and polarization images (M7). M7 indicates that amyloid deposits on Congo red-stained slides typically appear pink-salmon under a brightfield microscope, while the same areas exhibit apple-green birefringence under polarized light microscopy. Instances where pink-salmon areas do not show birefringence, or where apple-green birefringence occurs without the typical brightfield morphology, indicate inconsistency. We selected several representative FOVs from our training dataset to establish baseline scores for evaluation metrics M5 to M7 (see Supplementary Fig. 5). Note that these examples, along with their scores, were presented to the evaluating pathologists prior to their assessments to aid in their calibration and understanding of the scoring system.

For part 1 evaluation, small patches (1024 × 1024 pixels, each corresponding to ~166 × 166 µm²) are randomly cropped from histochemical and virtual images without any overlapping FOVs. Then, each image is randomly augmented in the following ways: original, left-right flip, top-bottom flip, and random rotation at 90, 180 and 270 degrees. A total of 163 images were randomly shuffled and sent to pathologists without labeling. Pathologists scored four metrics evaluating the brightfield Congo red image quality: stain quality of nuclei, stain quality of cytoplasm, stain quality of extracellular space and stain contrast of congophilic areas. For part 2, we selected 56 large image patches (2048 × 2048 pixels, each corresponding to ~332 × 332 µm²), and collected the brightfield and birefringence images of the same FOVs. For each image bundle, we collected both histochemically and virtually stained images, and randomly augmented the image bundles while ensuring a different augmentation method for each, resulting in a total of 112 image bundles to be scored. Pathologists scored three metrics evaluating the birefringence channel: amyloid quantification (% of positive areas / total tissue surface), birefringence image quality, and inconsistency between brightfield and polarization Congo red images. For part 2, the evaluation of the bundled images, we customized the Napari (0.4.18) viewer[67] as a graphical user interface (GUI) for pathologists to conveniently evaluate the high-resolution images. The interface features keyboard bindings for switching between brightfield and birefringence views, selecting images, and controlling zoom functions for regions of interest using a mouse. All the participating pathologists were given a short tutorial on using the GUI, which included scored example FOVs selected from the training dataset. In both evaluations, any images that pathologists declined to give a score are excluded from the analysis.

## Statistics and reproducibility

All the network inferences were deterministic for given input image data except for negligible numerical variations in implementation. For pathologists' blind evaluations, the sample image patch locations were manually selected to maximize the number of sample patches with variable amounts of amyloid deposits while maintaining acceptable overlap regions. Randomization was done through augmentation as described in the Methods section and the pathologists were blinded at the time of evaluation to whether the sample was histochemically or virtually stained. Patches without a morphological appearance indicative of amyloidosis were not included.

## Reporting summary

Further information on research design is available in the Nature Portfolio Reporting Summary linked to this article.

## Data availability

The authors declare that all data supporting the results of this study are available within the main text and the Supplementary Information. The raw imaging dataset collected by the authors cannot be shared publicly due to IRB restrictions, under protocol IRB #HS-20-00151. Example testing images and model pretrained weights are available together with the code at: https://github.com/Mikeeeeeey/AmylVirtualStain[68]. Source data are provided with this paper.

## Code availability

Deep learning models reported in this work used standard libraries and scripts that are publicly available in PyTorch. The codes for our virtual staining framework can be found at: https://github.com/Mikeeeeeey/AmylVirtualStain[68].

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

## Acknowledgements

The Ozcan Research Group at UCLA acknowledges the support of the NSF Biophotonics Program and NIH P41, The National Center for Interventional Biophotonic Technologies. N.P. is partially supported by a PhRMA Foundation Translational Medicine Postdoctoral Fellowship. M.A. is partially supported by the Scientific and Technological Research Council of Turkiye (TUBITAK).

## Author contributions

A.O. and N.P. conceived the research, X.Y., M.A., B.B. P.C.C., and N.P. imaged the unlabeled tissue sections and conducted polarization imaging, B.B., X.Y. and G.Z. developed the image processing pipeline and prepared the dataset, W.D.W. assisted with obtaining the unlabeled tissue sections, Y.Z., X.Y., B.B., Y.L., and S.Y.S. trained the neural networks. N.P., K.A., and G.A.F. performed the staining quality evaluation and cellularity assessment on the virtual and histochemical stained images. X.Y., B.B., and Y.L. performed the result analysis and statistical study. X.Y., N.P., Y.Z., and B.B. prepared the manuscript, and all authors contributed to the manuscript. A.O. supervised the research.

## Competing interests

A.O. is the co-founder of a company (Pictor Labs) that commercializes virtual staining technology. The remaining authors declare no competing interests.
