## [Peer Review File · Nature Communications]

Reviewers' Comments:

Reviewer #1:

Remarks to the Author:

This paper presents the first demonstration of virtual staining to improve visualization of amyloid deposits. Leveraging deep learning-based techniques, the authors present a model that simultaneously learns the image transformation from unstained, autofluorescence microscopy images into virtually stained birefringence (polarization) and brightfield images with Congo-red staining appearance.

The authors validated their approach through pathologist non-inferiority evaluations of a number of metrics that were specific to histological appearance as well as amyloid quantification and birefringence appearance. Additionally, the authors also investigate the pathologist's scores on paired image FOVs and demonstrated that the resulting virtual autofluorescence to birefringence and autofluorescence to brightfield Congo-red stain were successful.

Because the paper is very well written, and the clinical value of virtual Congo-red staining and polarization imaging is of great importance, the reviewer believes this work should be published in Nature Communications. Nevertheless, the authors should attend to the following critiques.

1. The birefringence of the resulting virtual staining was only evaluated through a qualitative approach, where the pathologists assign scores from a scale of 1 to 4 for each of the FOV. Since registration of the virtual staining and histology image is available, can you evaluate each image quantitatively, on its birefringence value, on a per pixel level? Please include a color bar for birefringence images if possible.
2. Similarly, the amyloid quantification metric can be evaluated quantitatively as well. The author emphasized on the importance of accurate amyloid quantification to prevent false diagnoses. The author should demonstrate that the quantitative results (positive area over the tissue region) is the same in virtually stained images as the histology ground truth.
3. As discussed in the Results and shown in Fig 3b, M1-M3 received higher scores for histochemical images. Can you speculate as to why and what this means?
4. Figure 4 (c), some columns (samples) received higher score for the virtual staining for all three metrics (M5-7) while other received lower or similar score when compared to the histologically stained images. Please include example images that show both cases.
5. It is unclear what M7, the consistency between brightfield and polarization images, refers to. Please provide an explanation and include example images where the inconsistent features are pointed out by arrows.
6. The authors mention that the quality of the light microscope and polarizers can

influence detection ability of amyloid detection. One can imagine the same is true for autofluorescence (AF): the quality of the microscope will affect the input image, which may affect its performance in the virtual staining process. Did the authors do any augmentations to validate or explore how quality of the AF data affects the results? Important variables would likely include variability in the light source intensity and spectral profile, choice of exposure time, quality of the camera and its spectral response, as well as the choice of AF filters and their quality. This should be mentioned, along with a general sense of the added cost associated with transitioning from current equipment to the new equipment required for this process (i.e., availability of autofluorescence microscopes and their cost compared to polarization scopes), which not an insignificant hurdle in clinical adoption.

7. No rationale is given for the choice of autofluorescence filters used. It would be nice to have a biological basis for choosing the filters used. Were all four actually necessary? What information do you assume each filter set brings to the problem and how does that match with the known targets of Congo red staining and birefringence?

8. During pathologist review of augmented images, did they review multiple versions of the same image within their dataset? If so, can you report on the image-to-image agreement of pathologist ratings?

9. It would aid the reader if you could point out features consistent with amyloid deposits and not for Figure 2. Given the visible presence of discrepancies in the image (absent current quantitative analysis of the image similarity), it is hard to assess the true degree of agreement.

10. Why were different exposure times used for the different filter sets?

Reviewer #2:

Remarks to the Author:

Overall, this work demonstrates an application of Virtual Staining to a new modality, birefringence imaging, as well as a new target. (amyloid buildup in cardiac tissue) This is a valuable contribution to this line of work. The technique allows training through a relatively small amount of data, and the application of a co-learned registration technique is intriguing. The authors also provide strong background and argument for the clinical utility of a VS application in this target. My largest concern is with the evaluation of the stains. I think there is some low-hanging fruit the authors could add to the analysis to greatly improve their demonstration of the validity of the stains. In addition I provide a small laundry list of suggestions, corrections, and desiderata.

The authors miss what I consider to be 3 important virtual staining citations that would help inform the manuscript. Some of the work the authors have done may be improvements as well, and these works are a useful comparison point that I will refer to

in this review. Citations at bottom.

1. Evaluation. The authors note (correctly) that generative neural nets are capable of hallucinating visual features. They suggest, in my professional opinion, incorrectly, that their choice of a paired-image translation framework as opposed to CycleGAN alleviates the risk of this. In my experience, feature hallucination is still a substantial risk, and as such, it is crucial to evaluate images by some metrics that inform us as to whether biological features are preserved between real and virtual test images. The metric analysis the authors provide, all of which are stain quality metrics, are indeed valuable. But while this analysis demonstrates that overall quality of the stain is preserved, it does not well-demonstrate that virtual stains preserve biological information. Namely, what if the network generates a high-quality virtual stain of imagined tissue? A simple example is nuclear dropout. [2] provides some off-the-shelf computational metrics that can be used to test image parity, and [1] shows parity of clinical evaluations made off of specific slides. Consider: Segmentation model result parity, nuclear segmentation algorithms, color spectrum analysis, FID, L1 loss between paired images, etc. Demonstrating parity of some kind of clinical judgment would be even better, but I understand it may not be possible at this point in the research program.

2. Registration Module

This is an interesting approach, and one which could be a potential improvement on the shift-invariant loss presented in [1]. Why was it trained simultaneously with the VS network? One could imagine learning a registration function using simple data augmentation, where the ground -truth displacement is known absolutely. This is a point that could be highlighted in the discussion, as I think the registration module itself is a useful contribution to the VS literature.

3. There is some inconsistency in the reported set of patches used for testing. Near the start of the Results section, it is claimed to be 163 at 1024x1024 resolution. Later, I see 65 at 2048x2048 resolution. Could it be made clear which sets are used for which evaluations? In addition please confirm that the set of 65 all belong to the 2 held-out patients.

4. In Supplementary Figure 2, the quality assessment of the virtual vs histochemical brightfield-stained images is shown. Why is this not shown on corresponding patches? My understanding of the experiment protocol suggests that this should be possible, and it is done for the birefringence images. If there is some reason that per-sample comparison could not be done for the brightfield images, the authors should clarify why. I do understand that the birefringence result is more important for the impact of the

work. See also Figure 3 vs. Figure 4.

5. Mathematical notation suggestions. I found the notation to be slightly non-standard, at least using the generative image network ML literature as a reference point.:

- The Binary Cross Entropy should use p_a rather than a as the variable, as this denotes that the loss makes sense in situations where the inputs represent probabilities.
- Use \ln to represent the natural log function.

The use of curly braces to enclose function arguments was a little odd.

- \times means cross product. Use implicit multiplication notation.
- Variables and constants should be italicized. Function calls should not be. (will help make things more readable for above bullet as well)
- Uppercase L is more often used to represent losses than l .

Just use $\|\dots\|_1$ for L_1 . This would make the equation visually cleaner.

6. Methods - Network Architecture and training strategy

- “where $[\cdot]$ represents the concatenation operation” seems like a broken sentence, as this operator is not used in the above or following formulas?
- “The coefficients (α, β, γ) in Eq. 2 were empirically set as $(10, 1, 0.0001)$.” Please confirm in the text that “empirically” means as a hyperparameter optimization based on training or evaluation set images, not on the images for which results are presented.

References:

[1] McNeil, Carson, Pok Fai Wong, Niranjan Sridhar, Yang Wang, Charles Santori, Cheng-Hsun Wu, Andrew Homyk, et al. 2023. “An End-to-End Platform for Digital Pathology Using Hyperspectral Autofluorescence Microscopy and Deep Learning Based Virtual Histology.” *Modern Pathology: An Official Journal of the United States and Canadian Academy of Pathology, Inc*, November, 100377.

[2] Zingman, I., Frayle, S., Tankoyeu, I., Sukhanov, S. & Heinemann, F.. (2024). A comparative evaluation of image-to-image translation methods for stain transfer in histopathology. *Medical Imaging with Deep Learning*, in *Proceedings of Machine Learning Research* 227:1509-1525 Available from <https://proceedings.mlr.press/v227/zingman24a.html>.

[3] Bayramoglu, Neslihan, Mika Kaakinen, Lauri Eklund, and Janne Heikkilä. 2017. “Towards Virtual H&E Staining of Hyperspectral Lung Histology Images Using

Conditional Generative Adversarial Networks.” In 2017 IEEE International Conference on Computer Vision Workshops (ICCVW), 64–71.

Reviewer #3:

Remarks to the Author:

Reviewer #4:

The team has embarked on a great journey to develop amyloid diagnosis without the usage of manual screening of the tissue for amyloid deposit. The effort made is extremely novel. However I have some general comments on the study, which must be discussed and experimentally proved.

1) The authors should make use of language at least at certain places where a layman (pathologist) or a researcher of biology will understand manuscript clearly. For example, what is computational staining to a layman pathologist, will bring clarity. The authors should find such areas of improvement for the readers of the manuscript.

2) While the authors in the abstract and the introduction have mention drawback of current followed methods as “However, Congo red staining is tedious and costly to perform, and prone to false diagnoses due to variations in the amount of amyloid, staining quality and expert interpretation through manual examination of tissue under a polarization microscope” I was expecting that all these issues would have resolved. For example, many methods development parameter which need to be talked about in a new method development should be considering e.g. accuracy of the method, precision of the method (repeatability, reproducibility, and intermediate precision), specificity, range of detection, linearity and limit detection, limit of quantitation. Since the authors have done some analysis on some of these parameters, it is utmost important to impress upon all these parameters, since they have the data in hand. Without addressing them will not give justification of such a novel method developed but without appreciating the main moto of developing this method.

3) The authors need to clarify to the less literate people about deep learning in the area what is taken as the input to develop virtual images and birefringence images. Is it that the starting model was developed based on multiple images chosen which were published in the literature or available as slides for Congo red staining in brightfield and polarizer microscopy. Was this applied to develop virtual images and then tested with the pathologists (this part is clear though). Otherwise, manuscript looks highly specialized only for deep learned people and not serve the purpose to the general audience.

4) By looking figure 2 and 3, it appears that virtual and stained images still vary in terms of intensity and colouring scheme (birefringence pattern).is these needs further improvement? needs to be commented.

5) There is a very important part of diagnosis of amyloids which goes one step above than having apple green birefringence, i.e. by changing the angle of polarizer to 10-degree, birefringence changes to pale green (clockwise), and yellow colour (anticlockwise). Please read and cite this paper (J Biosci 2021:46:14, Laboratory Investigation volume 88, pages232–242 (2008). I feel this feature was not included in the results. I suggest the authors to look or develop this feature to make their virtual method much closer to the Congo red staining

We sincerely thank the referees for their reviews and the constructive feedback that we have received on our manuscript “***Virtual birefringence imaging and histological staining of amyloid deposits in label-free tissue using autofluorescence microscopy and deep learning***” submitted to *Nature Communications* (Manuscript ID: NCOMMS-24-13210).

As detailed below, we have revised our manuscript in response to the reviewers’ comments. The original referee comments are shown in black color, whereas for ease of communication, our answers are provided in blue. Our revisions have also been marked in the main text and supplementary information files using yellow highlighting.

Summary of our revisions:

We have thoroughly revised our manuscript, incorporating several quantitative analyses and pathologist evaluations to support the success of our virtual staining method. Furthermore, we have revised our introduction and discussion sections and included a number of new figures in response to comments from the reviewers. The changes are detailed in specific responses below.

List of figure changes:

Revised Figure:

- **Figure 2.** Blind testing results of virtual birefringence imaging and histological staining of amyloid deposits in label-free whole slide images with zoomed-in regions also shown.

New Figures:

- **Figure 5.** Quantitative evaluation results for comparing the histochemically and virtually stained polarization Congo red images.
- **Supplementary Figure 1.** Congo Red Virtual Staining: Training and Testing.
- **Supplementary Figure 4.** Paired comparison of brightfield image quality.
- **Supplementary Figure 5.** Example fields-of-view with calibrated scores for M5-M7. All images are selected from the training dataset.
- **Supplementary Figure 8.** Additional examples of larger image patches and pathologist scores for birefringence image quality.
- **Supplementary Figure 9.** Workflow for calculating quantitative metrics used for the comparison of histochemically and virtually stained Congo-red images.
- **Supplementary Figure 10.** Quantitative evaluation results for comparing histochemically and virtually stained brightfield Congo-red images.
- **Supplementary Figure 11.** Transfer learning for a sensor with a higher noise level.
- **Supplementary Figure 12.** Schematic of virtual birefringence imaging with a shifted angle.
- **Supplementary Figure 13.** Virtual birefringence imaging with a shifted angle.
- **Supplementary Figure 14.** Ablation study with different numbers of input channels.

Reviewer #1:

This paper presents the first demonstration of virtual staining to improve visualization of amyloid deposits. Leveraging deep learning-based techniques, the authors present a model that simultaneously learns the image transformation from unstained, autofluorescence microscopy images into virtually stained birefringence (polarization) and brightfield images with Congo-red staining appearance.

The authors validated their approach through pathologist non-inferiority evaluations of a number of metrics that were specific to histological appearance as well as amyloid quantification and birefringence appearance. Additionally, the authors also investigate the pathologist’s scores on paired image FOVs and demonstrated that the resulting virtual autofluorescence to birefringence and autofluorescence to brightfield Congo-red stain were successful.

Because the paper is very well written, and the clinical value of virtual Congo-red staining and polarization imaging is of great importance, the reviewer believes this work should be published in Nature Communications. Nevertheless, the authors should attend to the following critiques.

We thank the reviewer for the positive feedback and constructive comments that helped us further enhance our manuscript.

- (1) The birefringence of the resulting virtual staining was only evaluated through a qualitative approach, where the pathologists assign scores from a scale of 1 to 4 for each of the FOV. Since registration of the virtual staining and histology image is available, can you evaluate each image quantitatively, on its birefringence value, on a per pixel level? Please include a color bar for birefringence images if possible.

We thank the reviewer for the constructive comments and valuable questions. In our revised manuscript, we've included several quantitative evaluations and added related analyses to the Results section ("Virtual staining model quantitative evaluation"); also see **Figure 5** and **Supplementary Figures 9-10**. The methods section was revised accordingly to elaborate on how these metrics were computed. Our revisions are quoted below, also highlighted in yellow in the revised manuscript file:

*"...To further validate our results, we employed several quantitative metrics (see the Methods section and **Supplementary Figure 9**) to assess the level of agreement between virtually and histochemically stained images. We initially focused on birefringence images due to their clinical significance in amyloidosis detection. Our analysis comprised two aspects: the area of the apple-green regions and the color distribution of the images; see **Figure 5**. The table in **Figure 5a** presents the image comparison metrics that we used: mean absolute error (MAE), multiscale structural similarity index metric (MS-SSIM), peak signal-to-noise ratio (PSNR), and Fréchet inception distance (FID). The metric values (low MAE and FID; high MS-SSIM and PSNR) and low standard deviations indicate a strong agreement between our virtually stained images and the corresponding histochemical ground truth. We also segmented the apple-green birefringent regions and measured the accuracy of our predictions using down-sampled intersection-over-union (D-IoU); see the Methods section for details. Two example FOVs with their segmented masks and the corresponding D-IoU values are shown in **Figure 5b**. **Figure 5c** also displays the D-IoU distribution of all the samples, highlighting the high concordance between the output images and the ground truth.*

*Next, we assessed the color distribution of (1) the segmented apple-green areas and (2) the entire images. We converted the virtual and histochemical images of two FOVs into YCbCr channels and plotted the histograms of each channel where the luminance information is stored as a single component (Y), and the chrominance information is stored as two color-difference components (Cb and Cr). The results, shown in **Figure 5d**, demonstrate a very good agreement in the color distribution, validating the color accuracy of our model inference.*

*For the brightfield Congo red images, we applied similar quantitative metrics (MAE, MS-SSIM, PSNR, and FID), as shown in **Supplementary Figure 10a**; we also included two additional metrics, the total count and the average size of nuclei in an image per FOV displayed in scatter plots in **Supplementary Figure 10b**. These results further affirm the concordance between our virtual staining results and the histochemically stained ground truth images, demonstrating the effectiveness of our method."*

*"...To quantitatively evaluate the performance of brightfield Congo red virtual staining, we organized 56 FOVs of virtually generated brightfield Congo red images together with their corresponding histochemically stained images for paired image comparison. Similar to refs.^{23,50} we first used standard metrics of MAE, MS-SSIM, PSNR, and FID, as well as some other customized metrics for brightfield images, including the number of nuclei per FOV, and average area of nuclei per FOV, as shown in **Supplementary Figure 9(a)**. MAE was defined as,*

$$MAE = \frac{1}{M \times N} \sum_m \sum_n |A(m, n) - B(m, n)| \quad (1)$$

where A, B represent histochemically and virtually stained brightfield Congo red images, respectively, m and n are the pixel indices, and M × N denotes the total number of pixels in each image.

The MS-SSIM⁵¹ evaluated the SSIM between two images at different spatial levels, where the number of scales was set to 6, and the weights applied to each scale were set to [0.05, 0.05, 0.1, 0.15, 0.2, 0.45].

The PSNR was calculated using,

$$PNSR = 10 \lg \left(\frac{\max(A)^2}{MSE} \right) \quad (2)$$

where A presents the ground truth image (histochemically stained), and MSE was defined as,

$$MSE = \frac{1}{M \times N} \sum_m \sum_n [A(m, n) - B(m, n)]^2 \quad (3)$$

The FID was calculated according to the definition reported in ref.⁵² using the default feature number.

As for the quantifications of nuclei properties, we adopted a stain deconvolution method⁵³ to first separate the channel corresponding to nuclei staining. Then, Otsu's thresholding⁵⁴ and a series of morphological operations, such as image dilation and erosion, were applied to the nuclei channel to obtain the segmented binary nuclei mask. The number of nuclei per FOV was defined as the number of connected components in the binary nuclei mask, and the average area of nuclei per FOV corresponds to the average area of connected components across the whole binary nuclei mask with a unit of pixel².

To perform the quantitative evaluation of polarization Congo red virtual staining, a paired comparison was conducted on 56 paired histochemically and virtually stained images with the same FOVs used in the brightfield image evaluation. We also used the metrics of MAE, MS-SSIM, PSNR, and FID as defined earlier. There were also some differences: (1) for MS-SSIM, the weights applied on each scale are [0.45, 0.2, 0.15, 0.1, 0.05, 0.05]; (2) for the calculation of FID , to adjust the image brightness, we transferred both the histochemically and virtually stained polarization images into YCbCr color space, multiplied the Y-channel values by 1.5, and then transferred the image back to the ordinary RGB color space. In addition, we developed a segmentation algorithm to extract the apple-green birefringence regions on the polarization Congo red images by empirically setting thresholds on H, S, V channels after transferring them into the HSV color space. Subsequently, we applied several morphological operations, such as image opening and image closing. As shown in Supplementary Figure 9(b), we used $D-IoU$ metric to measure the similarity between the segmented birefringence masks of the histochemically and virtually stained polarization Congo red images, defined as follows,

$$D - IoU = \frac{\sum_m \sum_n [A_{32}(m, n) * B_{32}(m, n)]}{\sum_m \sum_n [\min \{(A_{32}(m, n)) + (B_{32}(m, n)), 1\}]} \quad (4)$$

where A_{32} and B_{32} correspond to the $32 \times$ down-sampled (bilinear) version of the segmented birefringence masks for the histochemically and virtually stained polarization Congo red images, respectively, and m and n denote the pixel indices. Before computing $D-IoU$, A_{32} and B_{32} were binarized with a small threshold for logical operations. Compared to the traditional per-pixel definition of IoU , this down-sampled $D-IoU$ is more suitable to measure the similarity between binary masks and is resilient to small pixel misalignments⁵⁵. Finally, for the whole FOV and the apple-green birefringence regions only, we drew the color distributions using probability density functions fitted from histograms separately calculated in Y, Cb, and Cr channels to compare the color similarity between the histochemically and virtually stained polarization Congo red images.”

Finally, we did not include a color bar for birefringence images, as they were captured with a standard RGB image sensor.

- (2) Similarly, the amyloid quantification metric can be evaluated quantitatively as well. The author emphasized on the importance of accurate amyloid quantification to prevent false diagnoses. The author should demonstrate that the quantitative results (positive area over the tissue region) is the same in virtually stained images as the histology ground truth.

As addressed in our response to comment 1, the amyloid quantification was evaluated using a customized segmentation algorithm. The analysis methodology and results have been added to the revised manuscript, and a new figure (**Figure 5**) was included in our revised manuscript to summarize these findings along with the new **Supplementary Figures 9-10**. In addition to our responses to comment 1, the following text was added to our revised manuscript:

“... We initially focused on birefringence images due to their clinical significance in amyloidosis detection. Our analysis comprised two aspects: the area of the apple-green regions and the color distribution of the images; see **Figure 5**. The table in **Figure 5a** presents the image comparison metrics that we used: mean absolute error (MAE), multiscale structural similarity index metric (MS-SSIM), peak signal-to-noise ratio (PSNR), and Fréchet inception

*distance (FID). The metric values (low MAE and FID; high MS-SSIM and PSNR) and low standard deviations indicate a strong agreement between our virtually stained images and the corresponding histochemical ground truth. We also segmented the apple-green birefringent regions and measured the accuracy of our predictions using down-sampled intersection-over-union (D-IoU); see the Methods section for details. Two example FOVs with their segmented masks and the corresponding D-IoU values are shown in **Figure 5b**. **Figure 5c** also displays the D-IoU distribution of all the samples, highlighting the high concordance between the output images and the ground truth.”*

- (3) As discussed in the Results and shown in Fig 3b, M1-M3 received higher scores for histochemical images. Can you speculate as to why and what this means?

We thank the reviewer for the valuable question. We would like to point out that M5-M7 (polarization evaluation metrics) are of high clinical importance to amyloidosis detection, while brightfield evaluation (calculated via M1-M4) is used by pathologists to identify the specific regions of interest to inspect under polarized light microscopy. To address the referee’s comment, we have expanded our discussion section to include the trade-offs of using a digital staining matrix and some limitations of our work:

“...The virtual staining network’s performance was superior in the polarization Congo red channel and comparable in the brightfield channel, when benchmarked against the histochemical ground truth. This change in behaviour might be due to the trade-off introduced by the DSM: compared to training two distinct neural networks for brightfield Congo red and birefringence image generation, where an optimal model for each modality could be created, employing a DSM within a single neural network may result in a model that is optimal for one imaging modality but slightly suboptimal for the other. On the other hand, DSM inference provides better structural alignment between the output channels (as quantified by M7), whereas two separate network models might produce divergent artifacts and inconsistencies, potentially confusing pathologists during diagnosis; that is why, we utilized the presented architecture of the DSM-based multi-modal inference to bring consistency between virtually generated polarization and brightfield channels.

A limiting factor in our study was the scarcity and limited size of cardiac biopsy samples that were available to us. We trained and tested our virtual Congo red staining on tissue slides obtained from a single pathology laboratory, which may not exhaustively represent all clinically significant features observable in cardiac amyloidosis samples. In future work, we plan to evaluate the performance of our presented approach on other tissue types, such as kidney, liver, and spleen, using tissue slides from additional histology laboratories, which will further enhance the generalization and inference performance of our model.”

- (4) Figure 4 (c), some columns (samples) received higher score for the virtual staining for all three metrics (M5-7) while other received lower or similar score when compared to the histologically stained images. Please include example images that show both cases.

We thank the reviewer for the suggestion. We have included additional images (**Supplementary Figure 8**) with different fields of view in which the virtual stained images received equal and lower scores.

*“...Additional image FOVs can be found in **Supplementary Figure 7**, and two cases where the virtually stained images received lower and equal scores are reported in **Supplementary Figure 8**.”*

- (5) It is unclear what M7, the consistency between brightfield and polarization images, refers to. Please provide an explanation and include example images where the inconsistent features are pointed out by arrows.

Following the referee’s suggestion, we have revised and expanded the methods section to better explain the metrics utilized for the M7 evaluation. Additionally, we included four exemplary FOVs along with their evaluation scores given by a pathologist. These can be found in **Supplementary Figure 5**. Quoted from our revised manuscript, here are our responses:

“...Three board-certified pathologists were included to blindly evaluate the image quality of histochemically stained and virtually stained cardiac tissue sections. The blind evaluations were conducted in two ways: (1) Brightfield Congo red stain image quality on small image patches with M1-M4, which are standard quantification metrics for histology images; and (2) Large patch bundled images with both brightfield and birefringence images of the same FOV. In anatomic pathology, the evaluation of Congo red-stained slides under polarized light microscopy yields a

*binary result—positive or negative for amyloidosis. To enhance the quantification of our virtual polarization model, we incorporated additional metrics: amyloid quantification (M5), the quality of birefringence appearance (M6), and the consistency between brightfield and polarization images (M7). M7 indicates that amyloid deposits on Congo red-stained slides typically appear pink-salmon under a brightfield microscope, while the same areas exhibit apple-green birefringence under polarized light microscopy. Instances where pink-salmon areas do not show birefringence, or where apple-green birefringence occurs without the typical brightfield morphology, indicate inconsistency. We selected several representative FOVs from our training dataset to establish baseline scores for evaluation metrics M5 to M7 (see **Supplementary Figure 5**). Note that these examples, along with their scores, were presented to the evaluating pathologists prior to their assessments to aid in their calibration and understanding of the scoring system.”*

- (6) The authors mention that the quality of the light microscope and polarizers can influence detection ability of amyloid detection. One can imagine the same is true for autofluorescence (AF): the quality of the microscope will affect the input image, which may affect its performance in the virtual staining process. Did the authors do any augmentations to validate or explore how quality of the AF data affects the results? Important variables would likely include variability in the light source intensity and spectral profile, choice of exposure time, quality of the camera and its spectral response, as well as the choice of AF filters and their quality. This should be mentioned, along with a general sense of the added cost associated with transitioning from current equipment to the new equipment required for this process (i.e., availability of autofluorescence microscopes and their cost compared to polarization scopes), which not an insignificant hurdle in clinical adoption.

We thank the reviewer for this important comment. We would like to emphasize that our system utilizes standard microscopy components and filters that are compatible with all the commercial digital slide scanners (e.g., Leica, Zeiss, Hamamatsu, Olympus, and Philips, etc.), facilitating seamless integration to existing digital pathology pipelines. Following the reviewer's suggestion, we have conducted additional analyses (illustrated in **Supplementary Figure 11**) to demonstrate our model's adaptability to noise in the image capture. The manuscript was revised to include these additional analyses, highlighting the adaptability of our method:

*“...To demonstrate our model's ability to handle variations in image signal-to-noise (SNR), we utilized transfer learning to bring resilience against noise. The results of this analysis are summarized in **Supplementary Figure 11** where we digitally added Gaussian noise to simulate autofluorescence images captured with lower SNR image sensors. Direct inference using those noisy autofluorescence images as input to a model trained on original images (before the addition of Gaussian noise) generated hallucinated output images that are not acceptable. However, after applying transfer learning to the original model using noise-added images, the model quickly learned to adapt to the noisy inputs and inferred stained images that closely match the histochemical ground truth images (see **Supplementary Figure 11**). This showcases that our virtual staining model can be rapidly fine-tuned through transfer learning to effectively adapt to new configurations and maintain high fidelity.”*

- (7) No rationale is given for the choice of autofluorescence filters used. It would be nice to have a biological basis for choosing the filters used. Were all four actually necessary? What information do you assume each filter set brings to the problem and how does that match with the known targets of Congo red staining and birefringence?

We thank the reviewer for the insightful question. To explore the significance of the number of input autofluorescence channels in our study, we conducted an ablation study. For this analysis, we progressively reduced the number of input autofluorescence channels. The results of this study are detailed in a new supplementary figure (**Supplementary Figure 14**). The findings clearly indicate that the performance of our virtual staining algorithm decreases with fewer input channels. Regarding the rationale behind our choice of filters: previous virtual staining studies have utilized standard fluorescence filter sets that are commonly available on commercial slide scanners to better facilitate acceptance by digital pathology workflow. To the best of our knowledge, there are no studies directly examining the correlation between autofluorescence signals and Congo red staining targets alongside birefringence signals.

Based on the reviewer's comment, we revised the manuscript as follows:

*“...We also conducted an ablation study (see **Supplementary Figure 14**) to show that with the decreasing number of input autofluorescence channels, the output image quality of the virtual staining network decreases. This indicated that four input autofluorescence channels (DAPI, FITC, TxRed and Cy5) are indeed essential to acquire the necessary information for successfully generating the virtually stained images of amyloid deposits.”*

“...Note that the selection of the filters was primarily empirical, based on the availability of commonly used fluorescence filter sets, which spanned the visible spectrum while having spectral selectivity to capture distinct autofluorescence bands effectively”

(8) During pathologist review of augmented images, did they review multiple versions of the same image within their dataset? If so, can you report on the image-to-image agreement of pathologist ratings?

We thank the reviewer for the question. We did not include multiple versions of the same image with different augmentations to avoid potential memorization by evaluating pathologists that could introduce bias in our results.

(9) It would aid the reader if you could point out features consistent with amyloid deposits and not for Figure 2. Given the visible presence of discrepancies in the image (absent current quantitative analysis of the image similarity), it is hard to assess the true degree of agreement.

As detailed in our responses to comments 1 and 2, we've included quantitative analysis supporting the high degree of agreement between the two approaches (virtual vs. histochemical). In addition, we have updated Figure 2 to include additional annotations that enhance the interpretability of the images.

*“...The upward-pointing arrows with a blue outline highlight amyloid deposition between cardiac myocytes, while the right-pointing, yellow-outlined arrows highlight areas devoid of amyloid deposits. In **Figure 2(b)**, a similar comparative analysis is shown for another patient. In this case, the histochemically stained images display a specific region, labelled as ROI3, which is an area without any amyloid deposits (“negative region”). No congophilic features can be identified in the brightfield image, and no apple-green birefringence is seen in the polarization channel. The upward-pointing, orange-outlined arrows denote amyloid deposition within blood vessels. As indicated by the WSIs, zoomed-in regions and arrow-pointed specific areas, our virtual staining model correctly replicated these morphological characteristics, aligning closely with the histochemically stained ground truth images, without introducing false positive staining.”*

These modifications help to illustrate the success of the virtual staining technique. The marked regions show that the output from the virtually stained images closely aligns with the traditional histochemically stained images, affirming their clinical relevance.

(10) Why were different exposure times used for the different filter sets?

We have revised the methods section to explain the selection of the exposure times:

“...The exposure time for each channel was selected based on the intensity levels observed, which helped us to use a substantial portion of the dynamic range of the image sensor.”

Remark on code: I did not try to install and run the code but merely reviewed it to gauge readability. In general, I feel the code is not well documented, which may make it somewhat harder for someone who is not an expert to get started applying and testing this work.

- We thank the reviewer for the comment. In response, we have updated our GitHub repository's README page to include guidelines that are more user-friendly and accessible to non-experts.

Reviewer #2:

Overall, this work demonstrates an application of Virtual Staining to a new modality, birefringence imaging, as well as a new target. (amyloid buildup in cardiac tissue) This is a valuable contribution to this line of work. The technique allows training through a relatively small amount of data, and the application of a co-learned registration technique is intriguing. The authors also provide strong background and argument for the clinical utility of a VS application in this target. My largest concern is with the evaluation of the stains. I think there is some low-hanging fruit the authors could add to the analysis to greatly improve their demonstration of the validity of the stains. In addition I provide a small laundry list of suggestions, corrections, and desiderata.

We thank the reviewer for their positive feedback and suggestions, which have helped us to further enhance our manuscript and improve clarity. Point-by-point responses are included below.

- (1) Evaluation. The authors note (correctly) that generative neural nets are capable of hallucinating visual features. They suggest, in my professional opinion, incorrectly, that their choice of a paired-image translation framework as opposed to CycleGAN alleviates the risk of this. In my experience, feature hallucination is still a substantial risk, and as such, it is crucial to evaluate images by some metrics that inform us as to whether biological features are preserved between real and virtual test images. The metric analysis the authors provide, all of which are stain quality metrics, are indeed valuable. But while this analysis demonstrates that overall quality of the stain is preserved, it does not well-demonstrate that virtual stains preserve biological information. Namely, what if the network generates a high-quality virtual stain of imagined tissue? A simple example is nuclear dropout. [2] provides some off-the-shelf computational metrics that can be used to test image parity, and [1] shows parity of clinical evaluations made off of specific slides. Consider: Segmentation model result parity, nuclear segmentation algorithms, color spectrum analysis, FID, L1 loss between paired images, etc. Demonstrating parity of some kind of clinical judgment would be even better, but I understand it may not be possible at this point in the research program.

We thank the reviewer for the constructive comments and questions. In our revised manuscript, we've included several quantitative evaluations and added related analyses to the Results section ("Virtual staining model quantitative evaluation"); also see **Figure 5** and **Supplementary Figures 9-10**. The methods section was revised accordingly to elaborate on how these metrics were computed. Our revisions are quoted below, also highlighted in yellow in the revised manuscript file:

*"...To further validate our results, we employed several quantitative metrics (see the Methods section and **Supplementary Figure 9**) to assess the level of agreement between virtually and histochemically stained images. We initially focused on birefringence images due to their clinical significance in amyloidosis detection. Our analysis comprised two aspects: the area of the apple-green regions and the color distribution of the images; see **Figure 5**. The table in **Figure 5a** presents the image comparison metrics that we used: mean absolute error (MAE), multiscale structural similarity index metric (MS-SSIM), peak signal-to-noise ratio (PSNR), and Fréchet inception distance (FID). The metric values (low MAE and FID; high MS-SSIM and PSNR) and low standard deviations indicate a strong agreement between our virtually stained images and the corresponding histochemical ground truth. We also segmented the apple-green birefringent regions and measured the accuracy of our predictions using down-sampled intersection-over-union (D-IoU); see the Methods section for details. Two example FOVs with their segmented masks and the corresponding D-IoU values are shown in **Figure 5b**. **Figure 5c** also displays the D-IoU distribution of all the samples, highlighting the high concordance between the output images and the ground truth.*

*Next, we assessed the color distribution of (1) the segmented apple-green areas and (2) the entire images. We converted the virtual and histochemical images of two FOVs into YCbCr channels and plotted the histograms of each channel where the luminance information is stored as a single component (Y), and the chrominance information is stored as two color-difference components (Cb and Cr). The results, shown in **Figure 5d**, demonstrate a very good agreement in the color distribution, validating the color accuracy of our model inference.*

*For the brightfield Congo red images, we applied similar quantitative metrics (MAE, MS-SSIM, PSNR, and FID), as shown in **Supplementary Figure 10a**; we also included two additional metrics, the total count and the average size of nuclei in an image per FOV displayed in scatter plots in **Supplementary Figure 10b**. These results further affirm the concordance between our virtual staining results and the histochemically stained ground truth images, demonstrating the effectiveness of our method."*

*"...To quantitatively evaluate the performance of brightfield Congo red virtual staining, we organized 56 FOVs of virtually generated brightfield Congo red images together with their corresponding histochemically stained images for paired image comparison. Similar to refs.^{23,50} we first used standard metrics of MAE, MS-SSIM, PSNR, and FID, as well as some other customized metrics for brightfield images, including the number of nuclei per FOV, and average area of nuclei per FOV, as shown in **Supplementary Figure 9(a)**. MAE was defined as,*

$$MAE = \frac{1}{M \times N} \sum_m \sum_n |A(m, n) - B(m, n)| \quad (1)$$

where A, B represent histochemically and virtually stained brightfield Congo red images, respectively, m and n are the pixel indices, and M × N denotes the total number of pixels in each image.

The MS-SSIM⁵¹ evaluated the SSIM between two images at different spatial levels, where the number of scales was set to 6, and the weights applied to each scale were set to [0.05, 0.05, 0.1, 0.15, 0.2, 0.45].

The PSNR was calculated using,

$$PNSR = 10 \lg \left(\frac{\max(A)^2}{MSE} \right) \quad (2)$$

where A presents the ground truth image (histochemically stained), and MSE was defined as,

$$MSE = \frac{1}{M \times N} \sum_m \sum_n [A(m, n) - B(m, n)]^2 \quad (3)$$

The FID was calculated according to the definition reported in ref.⁵² using the default feature number.

As for the quantifications of nuclei properties, we adopted a stain deconvolution method⁵³ to first separate the channel corresponding to nuclei staining. Then, Otsu's thresholding⁵⁴ and a series of morphological operations, such as image dilation and erosion, were applied to the nuclei channel to obtain the segmented binary nuclei mask. The number of nuclei per FOV was defined as the number of connected components in the binary nuclei mask, and the average area of nuclei per FOV corresponds to the average area of connected components across the whole binary nuclei mask with a unit of pixel^2 .

To perform the quantitative evaluation of polarization Congo red virtual staining, a paired comparison was conducted on 56 paired histochemically and virtually stained images with the same FOVs used in the brightfield image evaluation. We also used the metrics of MAE, MS-SSIM, PNSR, and FID as defined earlier. There were also some differences: (1) for MS-SSIM, the weights applied on each scale are [0.45, 0.2, 0.15, 0.1, 0.05, 0.05]; (2) for the calculation of FID , to adjust the image brightness, we transferred both the histochemically and virtually stained polarization images into YCbCr color space, multiplied the Y-channel values by 1.5, and then transferred the image back to the ordinary RGB color space. In addition, we developed a segmentation algorithm to extract the apple-green birefringence regions on the polarization Congo red images by empirically setting thresholds on H, S, V channels after transferring them into the HSV color space. Subsequently, we applied several morphological operations, such as image opening and image closing. As shown in Supplementary Figure 9(b), we used D -IoU metric to measure the similarity between the segmented birefringence masks of the histochemically and virtually stained polarization Congo red images, defined as follows,

$$D - IoU = \frac{\sum_m \sum_n [A_{32}(m, n) * B_{32}(m, n)]}{\sum_m \sum_n [\min \{ (A_{32}(m, n)) + (B_{32}(m, n)), 1 \}]} \quad (4)$$

where A_{32} and B_{32} correspond to the $32 \times$ down-sampled (bilinear) version of the segmented birefringence masks for the histochemically and virtually stained polarization Congo red images, respectively, and m and n denote the pixel indices. Before computing D -IoU, A_{32} and B_{32} were binarized with a small threshold for logical operations. Compared to the traditional per-pixel definition of IoU, this down-sampled D -IoU is more suitable to measure the similarity between binary masks and is resilient to small pixel misalignments⁵⁵. Finally, for the whole FOV and the apple-green birefringence regions only, we drew the color distributions using probability density functions fitted from histograms separately calculated in Y, Cb, and Cr channels to compare the color similarity between the histochemically and virtually stained polarization Congo red images.”

(2) The Registration Module

This is an interesting approach, and one which could be a potential improvement on the shift-invariant loss presented in [1]. Why was it trained simultaneously with the VS network? One could imagine learning a registration function using simple data augmentation, where the ground -truth displacement is known absolutely. This is a point that could be highlighted in the discussion, as I think the registration module itself is a useful contribution to the VS literature.

We thank the viewer for this comment regarding the registration module. To address the referee's comments, we've expanded and modified our discussion on this topic as follows:

“...Unlike other shift-invariant losses used in earlier work⁵⁹, the registration module in our work was trained simultaneously with the virtual staining model to learn the misalignments between the target and output images from the virtual staining network. These misalignments include not only static optical aberrations but also complex tissue distortions introduced during the histochemical staining process, which can vary from one tissue FOV to another. These distortions are difficult to characterize with a simple loss function or data augmentation techniques but can be effectively learned with an optimizable network module.”

- (3) There is some inconsistency in the reported set of patches used for testing. Near the start of the Results section, it is claimed to be 163 at 1024x1024 resolution. Later, I see 65 at 2048x2048 resolution. Could it be made clear which sets are used for which evaluations? In addition please confirm that the set of 65 all belong to the 2 held-out patients.

We thank the reviewer for the valuable comment. First, we confirm that all evaluations were performed on the testing set that was never used during training. Regarding the two different evaluation configurations, they were both used to serve distinct purposes. The smaller patches were used for brightfield for standard staining quality, while the larger patches were used for analysis of birefringence appearance and image quality.

Our methods section on this is quoted below:

“...For part 1 evaluation, small patches (1024 × 1024 pixels, each corresponding to ~166 × 166 μm²) are randomly cropped from histochemical and virtual images without any overlapping FOVs. Then, each image is randomly augmented in the following ways: original, left-right flip, top-bottom flip, and random rotation at 90, 180 and 270 degrees. A total of 163 images were randomly shuffled and sent to pathologists without labeling. Pathologists scored four metrics evaluating the brightfield Congo red image quality: stain quality of nuclei, stain quality of cytoplasm, stain quality of extracellular space and stain contrast of congophilic areas. For part 2, we selected 56 large image patches (2048 × 2048 pixels, each corresponding to ~332 × 332 μm²), and collected the brightfield and birefringence images of the same FOVs.”

- (4) In Supplementary Figure 2, the quality assessment of the virtual vs histochemical brightfield-stained images is shown. Why is this not shown on corresponding patches? My understanding of the experiment protocol suggests that this should be possible, and it is done for the birefringence images. If there is some reason that per-sample comparison could not be done for the brightfield images, the authors should clarify why. I do understand that the birefringence result is more important for the impact of the work. See also Figure 3 vs. Figure 4.

To address the referee's comments, we've added a paired comparison of the brightfield image quality score comparison in **Supplementary Figure 4**. The results section of our revised manuscript was updated accordingly:

*“...The brightfield images of paired histochemical and virtually stained FOVs were further scored after a two-month washout period. These results, summarized in **Supplementary Figure 4**, further support that the virtual brightfield images generated by our model remain satisfactory and consistent.”*

- (5) Mathematical notation suggestions. I found the notation to be slightly non-standard, at least using the generative image network ML literature as a reference point.:
- The Binary Cross Entropy should use p_a rather than a as the variable, as this denotes that the loss makes sense in situations where the inputs represent probabilities.
 - Use \ln to represent the natural log function.
- The use of curly braces to enclose function arguments was a little odd.
- \times means cross product. Use implicit multiplication notation.
 - Variables and constants should be italicized. Function calls should not be. (will help make things more readable for above bullet as well)
 - Uppercase L is more often used to represent losses than l .
- Just use $\| \cdot \|_1$ for L1. This would make the equation visually cleaner.

We've edited the methods section as suggested by the referee for improved clarity.

- (6) Methods - Network Architecture and training strategy
- “where $[\cdot]$ represents the concatenation operation” seems like a broken sentence, as this operator is not used in the above or following formulas?
 - “The coefficients (α, β, γ) in Eq. 2 were empirically set as (10,1, 0.0001).” Please confirm in the text that “empirically” means as a hyperparameter optimization based on training or evaluation set images, not on the images for which results are presented.

We thank the reviewer for pointing this out. We've revised the sentence and notation for coherence:

“...where \tilde{c} is concatenated to the input as an additional channel. $[c_1]$ and $[c_{-1}]$ denote matrices with all the elements equal to 1 and -1, respectively. Note that for the additional 3rd channel required for yellow-colored amyloid deposits (see Supplementary Fig. 12), Eq. 5 is expanded to $\tilde{c} = [c_{-1}]$ or $[c_1]$ or $[c_2]$.”

The hyperparameter is empirically selected by examining validation set performances and is not optimized over the test set; quoted below from our revised manuscript:

“The coefficients (α, β, γ) in Eq. 6 were empirically set as (10,1,0.0001) based on the validation image set and were fixed for the finalized model evaluation on test samples.”

References:

[1] McNeil, Carson, Pok Fai Wong, Niranjan Sridhar, Yang Wang, Charles Santori, Cheng-Hsun Wu, Andrew Homyk, et al. 2023. “An End-to-End Platform for Digital Pathology Using Hyperspectral Autofluorescence Microscopy and Deep Learning Based Virtual Histology.” *Modern Pathology: An Official Journal of the United States and Canadian Academy of Pathology, Inc*, November, 100377.

[2] Zingman, I., Frayle, S., Tankoyeu, I., Sukhanov, S. & Heinemann, F.. (2024). A comparative evaluation of image-to-image translation methods for stain transfer in histopathology. *Medical Imaging with Deep Learning*, in *Proceedings of Machine Learning Research* 227:1509-1525 Available from <https://proceedings.mlr.press/v227/zingman24a.html>.

[3] Bayramoglu, Neslihan, Mika Kaakinen, Lauri Eklund, and Janne Heikkilä. 2017. “Towards Virtual H&E Staining of Hyperspectral Lung Histology Images Using Conditional Generative Adversarial Networks.” In *2017 IEEE International Conference on Computer Vision Workshops (ICCVW)*, 64–71.

Reviewer #3:

We thank the co-reviewer for the constructive feedback and comments.

Reviewer #4:

The team has embarked on a great journey to develop amyloid diagnosis without the usage of manual screening of the tissue for amyloid deposit. The effort made is extremely novel. However I have some general comments on the study, which must be discussed and experimentally proved.

(1) The authors should make use of language at least at certain places where a layman(pathologist) or a researcher of biology will understand manuscript clearly. For example, what is computational staining to a layman pathologist, will bring clarity. The authors should find such areas of improvement for the readers of the manuscript.

This is a great suggestion made by the referee - thanks. We have accordingly modified our manuscript to enhance its accessibility to clinicians and biologists, as quoted below.

“...In this manuscript, we introduced a novel approach that combines virtual birefringence imaging and virtual Congo red staining, enabling label-free imaging and detection of amyloid deposits. Specifically, we leveraged tissue autofluorescence texture at the sub-micron scale to generate virtual images that mimic the Congo red-stained tissue brightfield and polarization images, facilitating accurate identification of amyloid deposits within label-free tissue.”

In addition, we've added a **new Supplementary Figure 1** to better illustrate our pipeline for clinicians and biologists. **Supplementary Figure 1** Caption:

“Congo Red Virtual Staining: Training and Testing. (a) In the traditional pathology workflow, unstained slides suspected of having amyloid deposits undergo Congo red staining. A pathologist examines the Congo red-stained

slides, and if pink-salmon areas are identified, these areas are visualized under polarized light microscopy to provide a definitive diagnosis (1→3→4). For the virtual Congo red training step, unstained slides undergo autofluorescence scanning followed by Congo red histochemical staining (the latter used as ground truth). The Congo red-stained slides are then scanned using brightfield and polarized microscopy. By training a deep neural network-based algorithm, our method learns to transform the unstained/label-free slide into a virtual Congo red stain, in both its brightfield and polarization image channels (1→2→3→4→5). (b) During the model evaluation phase, unstained slides are converted into virtual Congo red-stained slides and sent for pathologist assessment.”

- (2) While the authors in the abstract and the introduction have mention drawback of current followed methods as “However, Congo red staining is tedious and costly to perform, and prone to false diagnoses due to variations in the amount of amyloid, staining quality and expert interpretation through manual examination of tissue under a polarization microscope” I was expecting that all these issues would have resolved. For example, many methods development parameter which need to be talked about in a new method development should be considering e.g. accuracy of the method, precision of the method (repeatability, reproducibility, and intermediate precision), specificity, range of detection, linearity and limit detection, limit of quantitation. Since the authors have done some analysis on some of these parameters, it is utmost important to impress upon all these parameters, since they have the data in hand. Without addressing them will not give justification of such a novel method developed but without appreciating the main moto of developing this method.

We thank the reviewer for the constructive comments. To quantify the detection capabilities of our methods, we conducted new analyses to compare the amyloid deposit areas. We used an IoU-based metric to compare the amyloid deposit detection accuracy as it penalizes both false-positives and false-negatives. In addition, we also compared the color distribution accompanied with other standard image evaluation metrics to better support our conclusions.

Following the referee’s comments, in our revised manuscript, we’ve included several quantitative evaluations and added related analyses to the Results section (“Virtual staining model quantitative evaluation”); also see **Figure 5** and **Supplementary Figures 9-10**. The methods section was revised accordingly to elaborate on how these metrics were computed. Our revisions are quoted below, also highlighted in yellow in the revised manuscript file:

*“...To further validate our results, we employed several quantitative metrics (see the Methods section and **Supplementary Figure 9**) to assess the level of agreement between virtually and histochemically stained images. We initially focused on birefringence images due to their clinical significance in amyloidosis detection. Our analysis comprised two aspects: the area of the apple-green regions and the color distribution of the images; see **Figure 5**. The table in **Figure 5a** presents the image comparison metrics that we used: mean absolute error (MAE), multiscale structural similarity index metric (MS-SSIM), peak signal-to-noise ratio (PSNR), and Fréchet inception distance (FID). The metric values (low MAE and FID; high MS-SSIM and PSNR) and low standard deviations indicate a strong agreement between our virtually stained images and the corresponding histochemical ground truth. We also segmented the apple-green birefringent regions and measured the accuracy of our predictions using down-sampled intersection-over-union (D-IoU); see the Methods section for details. Two example FOVs with their segmented masks and the corresponding D-IoU values are shown in **Figure 5b**. **Figure 5c** also displays the D-IoU distribution of all the samples, highlighting the high concordance between the output images and the ground truth.*

*Next, we assessed the color distribution of (1) the segmented apple-green areas and (2) the entire images. We converted the virtual and histochemical images of two FOVs into YCbCr channels and plotted the histograms of each channel where the luminance information is stored as a single component (Y), and the chrominance information is stored as two color-difference components (Cb and Cr). The results, shown in **Figure 5d**, demonstrate a very good agreement in the color distribution, validating the color accuracy of our model inference.*

*For the brightfield Congo red images, we applied similar quantitative metrics (MAE, MS-SSIM, PSNR, and FID), as shown in **Supplementary Figure 10a**; we also included two additional metrics, the total count and the average size of nuclei in an image per FOV displayed in scatter plots in **Supplementary Figure 10b**. These results further affirm the concordance between our virtual staining results and the histochemically stained ground truth images, demonstrating the effectiveness of our method.”*

“...To quantitatively evaluate the performance of brightfield Congo red virtual staining, we organized 56 FOVs of virtually generated brightfield Congo red images together with their corresponding histochemically stained images

for paired image comparison. Similar to refs.^{23,50} we first used standard metrics of MAE, MS-SSIM, PSNR, and FID, as well as some other customized metrics for brightfield images, including the number of nuclei per FOV, and average area of nuclei per FOV, as shown in **Supplementary Figure 9(a)**. MAE was defined as,

$$MAE = \frac{1}{M \times N} \sum_m \sum_n |A(m, n) - B(m, n)| \quad (1)$$

where A, B represent histochemically and virtually stained brightfield Congo red images, respectively, m and n are the pixel indices, and $M \times N$ denotes the total number of pixels in each image.

The MS-SSIM⁵¹ evaluated the SSIM between two images at different spatial levels, where the number of scales was set to 6, and the weights applied to each scale were set to [0.05, 0.05, 0.1, 0.15, 0.2, 0.45].

The PSNR was calculated using,

$$PNSR = 10 \lg \left(\frac{\max(A)^2}{MSE} \right) \quad (2)$$

where A presents the ground truth image (histochemically stained), and MSE was defined as,

$$MSE = \frac{1}{M \times N} \sum_m \sum_n [A(m, n) - B(m, n)]^2 \quad (3)$$

The FID was calculated according to the definition reported in ref.⁵² using the default feature number.

As for the quantifications of nuclei properties, we adopted a stain deconvolution method⁵³ to first separate the channel corresponding to nuclei staining. Then, Otsu's thresholding⁵⁴ and a series of morphological operations, such as image dilation and erosion, were applied to the nuclei channel to obtain the segmented binary nuclei mask. The number of nuclei per FOV was defined as the number of connected components in the binary nuclei mask, and the average area of nuclei per FOV corresponds to the average area of connected components across the whole binary nuclei mask with a unit of pixel².

To perform the quantitative evaluation of polarization Congo red virtual staining, a paired comparison was conducted on 56 paired histochemically and virtually stained images with the same FOVs used in the brightfield image evaluation. We also used the metrics of MAE, MS-SSIM, PSNR, and FID as defined earlier. There were also some differences: (1) for MS-SSIM, the weights applied on each scale are [0.45, 0.2, 0.15, 0.1, 0.05, 0.05]; (2) for the calculation of FID, to adjust the image brightness, we transferred both the histochemically and virtually stained polarization images into YCbCr color space, multiplied the Y-channel values by 1.5, and then transferred the image back to the ordinary RGB color space. In addition, we developed a segmentation algorithm to extract the apple-green birefringence regions on the polarization Congo red images by empirically setting thresholds on H, S, V channels after transferring them into the HSV color space. Subsequently, we applied several morphological operations, such as image opening and image closing. As shown in Supplementary Figure 9(b), we used D-IoU metric to measure the similarity between the segmented birefringence masks of the histochemically and virtually stained polarization Congo red images, defined as follows,

$$D - IoU = \frac{\sum_m \sum_n [A_{32}(m, n) * B_{32}(m, n)]}{\sum_m \sum_n [\min \{ (A_{32}(m, n)) + (B_{32}(m, n)), 1 \}]} \quad (4)$$

where A_{32} and B_{32} correspond to the 32× down-sampled (bilinear) version of the segmented birefringence masks for the histochemically and virtually stained polarization Congo red images, respectively, and m and n denote the pixel indices. Before computing D-IoU, A_{32} and B_{32} were binarized with a small threshold for logical operations. Compared to the traditional per-pixel definition of IoU, this down-sampled D-IoU is more suitable to measure the similarity between binary masks and is resilient to small pixel misalignments⁵⁵. Finally, for the whole FOV and the apple-green birefringence regions only, we drew the color distributions using probability density functions fitted from histograms separately calculated in Y, Cb, and Cr channels to compare the color similarity between the histochemically and virtually stained polarization Congo red images.”

- (3) The authors need to clarify to the less literate people about deep learning in the area what is taken as the input to develop virtual images and birefringence images. Is it that the starting model was developed based on multiple images chosen which were published in the literature or available as slides for Congo red staining

in brightfield and polarizer microscopy. Was this applied to develop virtual images and then tested with the pathologists (this part is clear though). Otherwise, manuscript looks highly specialized only for deep learned people and not serve the purpose to the general audience.

We thank the reviewer for their comment. We've revised our introduction and discussion section to simplify the technical aspects and improve clarity for readers without expertise in deep learning.

In response to the reviewer's question, our model was trained and tested on WSIs that were captured in our lab, rather than on publicly available WSIs. Once the model training stage was completed, the model was fixed, and no additional fine-tuning was performed. Then, pathologists blindly evaluated the model's outputs and compared the virtual staining of unseen images to the images captured from the same slides after histochemical staining (the testing phase). Following the completion of the training and testing phases, this validated model can be applied for any new unstained WSI, to perform virtual Congo red staining.

In addition, we've added a **new Supplementary Figure 1** to better illustrate our pipeline for clinicians and biologists. **Supplementary Figure 1** Caption:

“Congo Red Virtual Staining: Training and Testing. (a) In the traditional pathology workflow, unstained slides suspected of having amyloid deposits undergo Congo red staining. A pathologist examines the Congo red-stained slides, and if pink-salmon areas are identified, these areas are visualized under polarized light microscopy to provide a definitive diagnosis (1→3→4). For the virtual Congo red training step, unstained slides undergo autofluorescence scanning followed by Congo red histochemical staining (the latter used as ground truth). The Congo red-stained slides are then scanned using brightfield and polarized microscopy. By training a deep neural network-based algorithm, our method learns to transform the unstained/label-free slide into a virtual Congo red stain, in both its brightfield and polarization image channels (1→2→3→4→5). (b) During the model evaluation phase, unstained slides are converted into virtual Congo red-stained slides and sent for pathologist assessment.”

(4) By looking figure 2 and 3, it appears that virtual and stained images still vary in terms of intensity and colouring scheme (birefringence pattern).is these needs further improvement? needs to be commented.

We thank the reviewer for the question. Our findings, extracted from the pathologist evaluations, suggest that despite some differences in those images, they convey equivalent clinically relevant information. Nonetheless, the model performance can still be improved. To address the referee's comment, we have expanded our discussion section to include some limitations of our work and future directions:

“...The virtual staining network's performance was superior in the polarization Congo red channel and comparable in the brightfield channel, when benchmarked against the histochemical ground truth. This change in behavior might be due to the trade-off introduced by the DSM: compared to training two distinct neural networks for brightfield Congo red and birefringence image generation, where an optimal model for each modality could be created, employing a DSM within a single neural network may result in a model that is optimal for one imaging modality but slightly suboptimal for the other. On the other hand, DSM inference provides better structural alignment between the output channels (as quantified by M7), whereas two separate network models might produce divergent artifacts and inconsistencies, potentially confusing pathologists during diagnosis; that is why, we utilized the presented architecture of the DSM-based multi-modal inference to bring consistency between virtually generated polarization and brightfield channels.

A limiting factor in our study was the scarcity and limited size of cardiac biopsy samples that were available to us. We trained and tested our virtual Congo red staining on tissue slides obtained from a single pathology laboratory, which may not exhaustively represent all clinically significant features observable in cardiac amyloidosis samples. In future work, we plan to evaluate the performance of our presented approach on other tissue types, such as kidney, liver, and spleen, using tissue slides from additional histology laboratories, which will further enhance the generalization and inference performance of our model.”

(5) There is a very important part of diagnosis of amyloids which goes one step above than having apple green birefringence, i.e. by changing the angle of polarizer to 10-degree, birefringence changes to pale green (clockwise), and yellow colour (anticlockwise). Please read and cite this paper (J Biosci 2021:46:14, Laboratory Investigation volume 88, pages232–242 (2008). I feel this feature was not included in the results.

I suggest the authors to look or develop this feature to make their virtual method much closer to the Congo red staining.

We thank the reviewer for the valuable and inspiring comment. We conducted a proof-of-concept analysis to visualize birefringence with a small polarizer rotation (see the **new Supplementary Figures 12 and 13**), demonstrating that our model can be fine-tuned to generate such images. These new analyses and supplementary figures are included in our revised manuscript, quoted below:

*“...In addition to cross-polarized imaging, our system can also enable pathologists to virtually rotate the polarization filters by a small angle. This introduces an additional capability to virtual Congo red staining and can be used to further validate the presence of amyloid deposits, which should change from an apple-green color to a yellow-like color with ~10 degrees rotation of the polarization filter³⁹. For its proof-of-concept, we demonstrated this capability by adding another channel in the DSM, where a pixel value of 2 in the staining matrix represents the angle-shifted birefringence channel, as shown in **Supplementary Figure 12**. This new neural network, with 3 different virtually stained output images in its inference, was then trained using digitally emulated images where the color of the amyloid deposits changed from apple-green to yellow (see the Methods section), as expected from a slight shift of the polarizer angle. The blind testing results of this new virtual staining network, shown in **Supplementary Figure 13**, indicate that it successfully generated birefringence output images where the amyloid deposits appear in yellow, in addition to generating the other two channels, i.e., the cross-polarized and brightfield images. These results serve as a proof-of-concept demonstration of the DSM's multiplexing capability and its potential for additional output channels to assist diagnosis by virtually transforming the polarization filter into different states, as desired.”*

To conclude, we sincerely thank the referees for their constructive comments and feedback, which helped us to further improve the quality and clarity of our manuscript.

We look forward to hearing back from you regarding our revised submission.

Reviewers' Comments:

Reviewer #1:

Remarks to the Author:

The authors have made significant effort to address the previous comments, particularly regarding the adaptability of the method and the lack of quantitative evaluations. They have calculated a number of quantitative metrics, such as image comparison metrics, segmentation metric D-IoU, color comparison metrics and nuclei-related metrics. The additional figures and texts in the methods section included in the revised manuscript help to further strengthen the conclusions of the paper and improve the readability.

I have several questions and comments regarding the added content that the authors could provide further clarifications.

1. Regarding the segmentation method. How were the HSV thresholds determined? Were they determined from all FOVs or a selection of the entire image set? What metrics were used to determine the selected threshold? Also, please justify the choice of HSV color space in this analysis.
2. Fig 5d. The YCbCr histograms are shown for two example image pairs, which showed excellent agreement. Could you also report the values for the entire dataset, similar to what you have done in Fig 5a?
3. Additionally, since there were slight differences in the segmentation masks between the histologically vs. virtually stained image, I wonder if you should report the YCbCr values using a union mask of the two.
4. Regarding the quantitative metrics, in general I think the authors could elaborate on the justification of why those chosen metrics are relevant and provide a discussion of what it means (from the perspective of the metrics) that the reported values indicate good agreement.
5. Supp fig 8. Example a1 and a2 seem to be up-side-down. Is this intentional?

Reviewer #2:

Remarks to the Author:

Thanks a ton to the authors for their comprehensive and rigorous updates. They certainly added a significant amount of analysis, and I am sure this took a lot of analytical and infrastructural work. I'm excited to see this version published.

I do remain a stickler for mathematical notation:

- In most places, "log" was replaced for the natural log "ln" as I recommended. Thank

you! However the PSNR formula in equation (2) uses "lg".

- There is a scattered use of different multiplication notations between implicit, " \times ", and " $*$ ". I will reiterate that " \times " should really be reserved for a vector cross-product and $*$ often means convolution in the ML literature. Please use implicit or multiplication notation when possible, and definitely use the same thing everywhere if making an exception to implicit. "XY" is preferable to " $X\times Y$ ". $X*Y$ is also better than $X\times Y$, but really, XY is best. See equations 1, 3, 4, 6, 7, 8, 10, 11, 12.

Reviewer #3:

Remarks to the Author:

Reviewer #4:

Remarks to the Author:

I am overall satisfied with the revisions, explanations and, additional work done by the authors.

However, I have small suggestions/corrections/

On page 11, the authors have used a very murky term, amyloid crystals. Why they have used it? what does it mean? If the authors are looking for synonym of fibre to write crystal, it is structurally not appropriate.

2) Collagen is a major component of any tissue and may give a birefringence/autofluorescence property, although not the way it gives with amyloid fibrils. Interferences with collagen should be commented or tested. Was this considered in the analysis. Sorry, I have missed this point during first revision.

3) I suggest to make supplementary figure 12 and 13 as a separate figure in the main text, as this characteristic feature is the most important feature of an amyloid fibril characterization. Also, as the analysis is done now, this will strengthen the paper quality.

We sincerely thank the referees for their reviews and the constructive feedback that we have received on our manuscript “***Virtual birefringence imaging and histological staining of amyloid deposits in label-free tissue using autofluorescence microscopy and deep learning***” submitted to *Nature Communications* (Manuscript ID: NCOMMS-24-13210A).

As detailed below, we have revised our manuscript in response to the reviewers’ comments. The original referee comments are shown in black color, whereas for ease of communication, our answers are provided in blue. Our revisions have also been marked in the main text and supplementary information files using yellow highlighting.

Summary of our revisions:

We’ve added an additional Supplementary Figure for color spectrum analysis, edited the manuscript for enhanced clarity and expanded the Discussion section in response to the reviewers’ comments. All of our changes are detailed in specific responses below.

Figure changes:

Figure 6. Virtual birefringence imaging with a shifted angle. (merged from Supplementary Figures 12 and 13 – moved into the main text, following a referee comment)

Supplementary Figure 10. Color histograms in YCbCr color space for the entire test dataset. (new figure)

Reviewer #1:

The authors have made significant effort to address the previous comments, particularly regarding the adaptability of the method and the lack of quantitative evaluations. They have calculated a number of quantitative metrics, such as image comparison metrics, segmentation metric D-IoU, color comparison metrics and nuclei-related metrics. The additional figures and texts in the methods section included in the revised manuscript help to further strengthen the conclusions of the paper and improve the readability. I have several questions and comments regarding the added content that the authors could provide further clarifications.

-- We thank the reviewer for the positive feedback and constructive comments that helped us further improve our manuscript.

1. Regarding the segmentation method. How were the HSV thresholds determined? Were they determined from all FOVs or a selection of the entire image set? What metrics were used to determine the selected threshold? Also, please justify the choice of HSV color space in this analysis.

-- We’ve clarified the segmentation method details in the Methods section, quoted below:

“...Additionally, we developed a segmentation algorithm to isolate the apple-green birefringence regions in polarization Congo red images by empirically setting thresholds on the HSV channels. The Hue channel in the HSV space enabled the direct selection of the green color range for amyloid deposits. Thresholds were determined from sample FOVs and then validated and finetuned by a board-certified pathologist to ensure accurate amyloid segmentation.”

2. Fig 5d. The YCbCr histograms are shown for two example image pairs, which showed excellent agreement. Could you also report the values for the entire dataset, similar to what you have done in Fig 5a?

-- We thank the reviewer for this suggestion. We have added a new **Supplementary Figure 10** to show the YCbCr color histogram comparisons for all test FOVs, and revised the Results section accordingly, quoted below:

*“...To further confirm the color accuracy, we calculated the pixel value distribution across the entire testing dataset and plotted the results in **Supplementary Figure 10**, which revealed a strong concordance across all the channels, as desired.”*

3. Additionally, since there were slight differences in the segmentation masks between the histologically vs. virtually stained image, I wonder if you should report the YCbCr values using a union mask of the two.

-- We do believe that using separate segmentation masks for histologically stained images and their virtually stained counterparts is a more accurate description of the amyloid deposit color distribution than employing a union mask. When extracting color distributions in target birefringence regions, if we were to apply a segmentation mask obtained from the virtually stained image to its corresponding histochemically stained counterpart, it would inadvertently include some background elements as interference, and vice versa. To ensure the YCbCr values measure only the birefringence regions without contamination from other background/tissue areas, using distinct segmentation masks for each staining method is a better approach, which worked very well.

4. Regarding the quantitative metrics, in general I think the authors could elaborate on the justification of why those chosen metrics are relevant and provide a discussion of what it means (from the perspective of the metrics) that the reported values indicate good agreement.

-- MSE, PSNR, and MS-SSIM are standard and commonly used metrics for assessing image differences. We selected MS-SSIM (instead of SSIM) because it evaluates pixel-wise agreement across different resolutions/magnifications, providing a more comprehensive assessment. These metrics show that our output images closely match the target images. FID is generally used for generative models to measure data distribution similarity. A low FID value represents that there is no significant difference in the distributions of the output and target images. The count and area of nuclei are standard pathological metrics, confirming that the brightfield images accurately reflect pathological features. The apple-green birefringence segmentation and color distribution (D-IoU and color histograms) are also important due to their high clinical relevance in diagnosing amyloidosis. A strong agreement in these areas indicates that our output images convey the same diagnostic information on amyloid deposits as the histochemically stained images.

We have revised the Results sections to further expand on these:

“Virtual staining model quantitative evaluation

To further validate our results, we employed several quantitative metrics (see the Methods section and Supplementary Figure 9) to assess the level of agreement between virtually and histochemically stained images. We initially focused on birefringence images due to their clinical significance in amyloidosis detection. Our analysis comprised two aspects: the area of the apple-green regions and the color distribution of the images; see Figure 5. The table in Figure 5a presents the image comparison metrics that we used: mean absolute error (MAE), multiscale structural similarity index metric (MS-SSIM), peak signal-to-noise ratio (PSNR), and Fréchet inception distance (FID). The metric values (low MAE and FID; high MS-SSIM and PSNR) and low standard deviations indicate a strong agreement between our virtually stained images and the corresponding histochemical ground truth. We also segmented the apple-green birefringent regions and measured the accuracy of our predictions using down-sampled intersection-over-union (D-IoU); see the Methods section for details. Two example FOVs with their segmented masks and the corresponding D-IoU values are shown in Figure 5b. Figure 5c also displays the D-IoU distribution of all the samples, highlighting the high concordance between the output images and the ground truth, where the regions containing amyloid deposits in the output images overlap significantly with those in the target images.

Next, we assessed the color distribution of (1) the segmented apple-green areas and (2) the entire images. We converted the virtual and histochemical images of two FOVs into YCbCr channels and plotted the histograms of each channel where the luminance information is stored as a single component (Y), and the chrominance information is stored as two color-difference components (Cb and Cr). The results, shown in Figure 5d, demonstrate a very good agreement in the color distribution, validating the color accuracy of our model inference. To further confirm the color accuracy, we calculated the pixel value distribution across the entire

testing dataset and plotted the results in Supplementary Figure 10, which revealed a strong concordance across all the channels, as desired.

For the brightfield Congo red images, we applied similar quantitative metrics (MAE, MS-SSIM, PSNR, and FID), as shown in Supplementary Figure 11a; we also included two additional metrics commonly used for brightfield images: the total count and the average size of nuclei in an image per FOV. The results are displayed in the scatter plots shown in Supplementary Figure 11b. These results further affirm the concordance between our virtual staining results and the histochemically stained ground truth brightfield images, demonstrating the effectiveness of our method.”

5. Supp fig 8. Example a1 and a2 seem to be up-side-down. Is this intentional?

-- We thank the reviewer for pointing this out. We've corrected the figure.

The code has few to no comments, suggesting documentation is poor. I generally find this unacceptable.

-- We have improved our comments for the uploaded codes for better readability and clarity.

Reviewer #2:

Thanks a ton to the authors for their comprehensive and rigorous updates. They certainly added a significant amount of analysis, and I am sure this took a lot of analytical and infrastructural work. I'm excited to see this version published.

I do remain a stickler for mathematical notation:

- In most places, "log" was replaced for the natural log "ln" as I recommended. Thank you! However the PSNR formula in equation (2) uses "lg".

- There is a scattered use of different multiplication notations between implicit, "x", and "*". I will reiterate that "x" should really be reserved for a vector cross-product and * often means convolution in the ML literature. Please use implicit or multiplication notation when possible, and definitely use the same thing everywhere if making an exception to implicit. "XY" is preferable to "XxY". X*Y is also better than XxY, but really, XY is best. See equations 1, 3, 4, 6, 7, 8, 10, 11, 12.

-- We thank the reviewer for the valuable comments and further remarks on the notations. We have edited our equations as suggested.

Reviewer #3:

-- We thank the co-reviewer for the constructive feedback and comments.

Reviewer #4:

I am overall satisfied with the revisions, explanations and, additional work done by the authors. However, I have small suggestions/corrections/

On page 11, the authors have used a very murky term, amyloid crystals. Why they have used it? what does it mean? If the authors are looking for synonym of fibre to write crystal, it is structurally not appropriate.

-- We thank the reviewer for pointing this out. We've changed "amyloid crystals" to "amyloid deposits" for clarity.

2) Collagen is a major component of any tissue and may give a birefringence/autofluorescence property, although not the way it gives with amyloid fibrils. Interferences with collagen should be commented or tested. Was this considered in the analysis. Sorry, I have missed this point during first revision.

-- We thank the reviewer for this valuable comment. To address this comment, we have added the following new text to our revised Discussion, quoted below:

"...For example, other tissue components like collagen can exhibit some birefringence that differs in appearance from amyloid fibrils. It is important to note that Congo red staining highlights amyloid deposits, and stains collagen/elastic fibers with far less affinity⁴⁰. Therefore, the typical apple-green birefringence signatures distinguish amyloid deposits from other fibrils and from the white birefringence of fibrin or collagen⁴¹. Moreover, since our virtual staining approach does not rely on birefringence images at its input channel and only uses autofluorescence images of label-free tissue, our approach should in principle be immune to various forms of non-specific tissue birefringence as long as their microscopic spatial features at multiple autofluorescence image channels do not substantially overlap with the spectral and spatial features of label-free amyloid autofluorescence. Thus, the fact that our input images utilize four different channels of autofluorescence (i.e., DAPI, FITC, TxRed and Cy5) helps with the specificity of our virtual staining approach for amyloid deposits. To further shed light on this, future work will involve validating our model's capability to accurately differentiate mimics of amyloidosis from amyloid deposits, based on their label-free autofluorescence images. We also plan to expand our evaluations to other tissue types, such as kidney, liver, and spleen, using tissue slides from additional histology laboratories, which will further enhance the generalization and inference performance of our model..."

3) I suggest to make supplementary figure 12 and 13 as a separate figure in the main text, as this characteristic feature is the most important feature of an amyloid fibril characterization. Also, as the analysis is done now, this will strengthen the paper quality.

-- Following the referee's suggestion, we have merged these two supplementary figures into the main text, Figure 6, and accordingly updated the manuscript.

To conclude, we sincerely thank the referees for their constructive comments and feedback, which helped us to further improve the quality and clarity of our manuscript.

We look forward to hearing back from you regarding our revised submission.

Reviewers' Comments:

Reviewer #1:

Remarks to the Author:

The author has sufficiently responded to previous comments.

Reviewer #2:

Remarks to the Author:

Thanks to the authors for their additional revisions. I recommend this article for publication.

Reviewer #3:

Remarks to the Author:

Reviewer #4:

Remarks to the Author:

I am satisfied with the author's responses and addition of text in the main manuscript.